# Tgfbr1 controls developmental plasticity between the hindlimb and external genitalia by remodeling their regulatory landscape

Anastasiia Lozovska [1], Artemis G. Korovesi [1], André Dias [1,2], Alexandre Lopes[1], Donald A. Fowler[1], Gabriel G. Martins [1], Ana Nóvoa [1] & Moisés Mallo [1] ✉

The hindlimb and external genitalia of present-day tetrapods are thought to derive from an ancestral common primordium that evolved to generate a wide diversity of structures adapted for efficient locomotion and mating in the ecological niche occupied by the species. We show that despite long evolutionary distance from the ancestral condition, the early primordium of the mouse external genitalia preserved the capacity to take hindlimb fates. In the absence of *Tgfbr1*, the pericloacal mesoderm generates an extra pair of hindlimbs at the expense of the external genitalia. It has been shown that the hindlimb and the genital primordia share many of their key regulatory factors. *Tgfbr1* controls the response to those factors by modulating the accessibility status of regulatory elements that control the gene regulatory networks leading to the formation of genital or hindlimb structures. Our work uncovers a remarkable tissue plasticity with potential implications in the evolution of the hindlimb/genital area of tetrapods, and identifies an additional mechanism for *Tgfbr1* activity that might also contribute to the control of other physiological or pathological processes.

The vertebrate body is built sequentially in a head to tail progression. While this is a continuous process, it involves two transitions entailing major changes in gene regulatory mechanisms. The first transition results in the switch from a head to a trunk developmental program[1], marking the start of axial extension and the layout of the primordia for most of the organ systems involved in vital and reproductive functions. The second transition organizes the end of trunk structures and activates the tail developmental program[2]. The transition from trunk to tail entails significant reorganization of embryonic structures involving derivatives from all germ layers. The neuromesodermal competent (NMC) progenitors[3,4] relocate from the epiblast to the tail bud, from where they keep extending the body axis[5,6]. The progenitors for the lateral plate mesoderm, involved in the formation and vascularization of the trunk-resident organs[7], undergo terminal differentiation leading to the formation of the hindlimbs and of the ventral lateral mesoderm that will then become the pericloacal mesoderm[2]. Reciprocal interactions between the pericloacal mesoderm and the endodermal cloaca will then generate the genital tubercle (GT), the precursor of the external genitalia, while also organizing the exit channels of the digestive and excretory systems with the formation of the rectus/anus, the bladder and the urethra[8–12]. Although the hindlimb shares many of the patterning and morphogenetic processes with the forelimb[13], genetic analyses indicate major regulatory differences controlling the earliest stages of development of the two appendages[14–16]. Interestingly, molecular studies revealed that the hindlimb and the GT share many of their regulatory processes[9,17–19], indicating that they might be closely connected developmentally. Indeed, it has been reported that in some tetrapod species hindlimbs and genitals share a common primordium, which might represent the ancestral condition[20]. In mammals, it has been suggested that the

[1]Instituto Gulbenkian de Ciência, Rua da Quinta Grande 6, 2780-156 Oeiras, Portugal. [2]Present address: Department of Experimental and Health Sciences, Universitat Pompeu Fabra, Barcelona, Spain. ✉e-mail: mallo@igc.gulbenkian.pt

posterior displacement of the cloaca relative to the hindlimb bud disconnected the limb primordium from the influence of the endodermal signals promoting genital fates, thus allowing formation of both legs and external genitalia[20].

Genetic studies, mostly in mouse embryos, revealed that the control of the early stages of hindlimb and cloacal/pericloacal development requires functional input from several regulatory factors. These include signaling pathways, like those of the transforming growth factor β/bone morphogenetic protein (Tgfβ/BMP) superfamily[21–25], fibroblast growth factors (FGFs)[26–28], sonic hedgehog (Shh)[29–31] and both canonical and non-canonical WNTs[10,11,32], as well as several transcription factors, including Hoxa13, Hoxd13[33,34] and Isl1[35]. Importantly, while the same regulatory factors play essential roles in both structures[9,18], their functional outcome differ substantially in the hindlimb and the GT, indicating the existence of mechanisms promoting distinct patterns of response in the hindlimb and the pericloacal mesoderm when exposed to a particular regulatory factor. Shh provides a paradigmatic example of such functional duality. It determines anterior/posterior limb bud polarity from the posterior limb bud mesenchyme regulating the identity of the limb skeletal elements[36]. Conversely, from its expression domain in the endodermal cloaca Shh controls the morphogenesis of the GT, as well as the organization of the different outlets for the urogenital and intestinal systems[29,30]. The mechanisms regulating this distinct tissue response to common regulatory factors are largely unknown.

Genetic analyses have shown that the Tgfβ receptor 1 (Tgfbr1) (also known as Alk5) plays a key role in the activation of the trunk to tail transition, also initiating the regulatory sequence controlling formation of the hindlimb and external genitalia[2,37,38]. Indeed, Tgfbr1 null mutant embryos fail to induce the primordia of the hindlimb or cloacal/pericloacal structures, as shown by the absence of early markers or morphological landmarks for these tissues[37]. Conversely, premature activation of Tgfbr1 signaling in the axial progenitors resulted in earlier induction of these structures, as estimated by gene marker expression and the substantial anterior displacement of the hindlimb buds[2].

It has been shown that the Growth and Differentiation factor 11 (Gdf11) is a physiological ligand for Tgfbr1 in its role at the trunk to tail transition[38]. Indeed, Gdf11 mouse mutant embryos show a significant delay in the transition to tail development[2,39]. Expression and functional analyses in several other vertebrates indicate that this role for Gdf11 is not restricted to the mouse but might be shared by most other tetrapods[40,41]. However, contrary to Tgfbr1 mutant embryos, Gdf11 mutants still activate the transition to tail development, although with a significant delay, indicating that other ligands in the Tgfβ/BMP family might cooperate with Gdf11 in triggering Tgfbr1 activity in axial tissues. One such factor is Gdf8 as revealed by the enhanced trunk size observed in Gdf11/Gdf8 compound mutants[42]. Bmp4 and Bmp7 also stand out as potential candidates to mediate some of the Tgfbr1 activities in the caudal body. Bmp7/Tsg, Bmp7/Shh mutants and compound Bmp7 homozygous/Bmp4 heterozygous conditional inactivation in the region of the trunk to tail transition resulted in the fusion of the hindlimbs (sirenomelia) due to the absence of ventral lateral mesoderm[22,23,43] and, thus, of the pericloacal mesoderm that builds the GT[20], which is also absent from Tgfbr1 null mutant embryos[21]. Although BMPs are not considered as canonical Tgfbr1 ligands[44], given the high complexity of interactions between different receptors of the Tgfβ/BMP superfamily[45–48], functional interactions between Tgfbr1 and Bmp4/Bmp7 cannot be ruled out, most particularly because the bona fide receptors for those BMPs in the pericloacal/hindlimb area await identification.

The possible involvement of different ligands of the Tgfβ/BMP superfamily in the activation of Tgfbr1 might indicate distinct regulation of Tgfbr1-dependent processes associated with the trunk to tail transition. We initially set to understand whether the proper regulation of NMC cell fate decisions after the transition that has been shown to require Gdf11 activity[49], is mediated by Tgfbr1 signaling. Given the early developmental lethality of Tgfbr1 null mutants derived from its role during early steps of heart development[50,51], we conditionally inactivated Tgfbr1 in the caudal embryonic region, thus bypassing its critical role in heart formation. These experiments, in addition to confirm the involvement of Tgfbr1 signaling in the control of NMC progenitors from the tail bud, resulted in a totally unpredicted phenotype, consisting in the development of a second set of hindlimbs originating from the pericloacal mesoderm, a tissue normally generating the external genitalia. By comparing the chromatin accessibility profiles of wild type and mutant tissues, we could observe that the mutant pericloacal mesoderm acquires a significant number of limb-type signatures, while losing many of those specifically associated with the GT. Our results indicate that Tgfbr1 modulates the type of response of the pericloacal mesoderm to patterning signals from the endoderm by remodeling the regulatory landscape of the chromatin, thus defining a limb or a GT-type of gene activation.

## Results

### Generation of Tgfbr1-cKO developmental model

For the conditional Tgfbr1 inactivation approach we used three Tgfbr1 alleles, namely the Tgfbr1 null allele[37] and either Tgfbr1$^{flox}$, carrying LoxP sites flanking Exon 3[52], or Tgfbr1$^{3ex3-flox}$, containing a triplicated Exon 3 each surrounded by LoxP sites (Supplementary Fig. 1). Characterization of mice homozygous for the Tgfbr1$^{3ex3-flox}$ allele revealed that this allele generates a transcript containing a tandem of three Exon 3 sequences in frame with the flanking Exons 2 and 4 (Supplementary Fig. 1C). Expression in HEK293T cells of the Tgfbr1$^{3ex3}$ mRNA isolated from the mouse embryos confirmed that it generates a protein product with the expected size of Tgfbr1 containing triplicated Exon 3 sequences (Supplementary Fig. 1D). Exon 3 encodes the transmembrane domain and major part of the GS domain of the protein, including the serin-rich region phosphorylated upon ligand binding and implicated in signal transduction[44,53]. Homozygous Tgfbr1$^{3ex3-flox}$ animals were not viable, and Tgfbr1$^{3ex3-flox/-}$ fetuses phenocopied the knock out of the Tgfbr1 ligand Gdf11[2,39] (Supplementary Fig. 1B, B'). Together, the above results strongly suggest that Tgfbr1$^{3ex3-flox}$ produces a hypomorphic receptor.

Tgfbr1 inactivation was promoted using the Cdx2Cre$^{ERT}$ transgenic driver[2], triggering Cre activity in caudal embryonic tissues by tamoxifen administration (Supplementary Fig. 1E). Experiments using either Tgfbr1 floxed allele gave similar phenotypes, but while Tgfbr1$^{flox/-}$;Cdx2Cre$^{ERT+/0}$ embryos required two consecutive tamoxifen doses at embryonic stage (E) 6.75 and E7.25, a single tamoxifen injection at E7.25 was sufficient to obtain the same mutant phenotype from for Tgfbr1$^{3ex3-flox/-}$;Cdx2Cre$^{ERT+/0}$ embryos, thus reducing the frequency of tamoxifen-derived late fetal miscarriage. Our tamoxifen treatment scheme inactivated Tgfbr1 in the tissues caudal from the forelimb bud, as confirmed by RT-qPCR analyses on RNA extracted from the posterior part of E10.5 embryos, including the hindlimbs (Supplementary Fig. 1F, G). Tamoxifen-treated embryos and fetuses with either the Tgfbr1$^{flox/-}$;Cdx2Cre$^{ERT+/0}$ or the Tgfbr1$^{3ex3-flox/-}$;Cdx2Cre$^{ERT+/0}$ genotype will be hereafter referred to as Tgfbr1-cKO.

### Tgfbr1-cKO exhibit multiple developmental malformations

The most prominent feature of Tgfbr1-cKO mutants was a duplication of their hindlimbs [4/5 at E16.5] (Fig. 1A, B). The fifth fetus analyzed at this stage lacked overt hindlimb duplication but still contained small protrusions posterior to the hindlimbs, a phenotype likely derived from less efficient recombination. The limb identity of the duplicated structures was confirmed by the presence of skeletal structures that, although variable in morphology, were clearly identified as belonging to limbs (Fig. 1C, Supplementary Fig. 2, and Supplementary Movie 1). For instance, a close look at one such duplicated hindlimb showed the presence of stylopod, zeugopod and autopod elements. When these

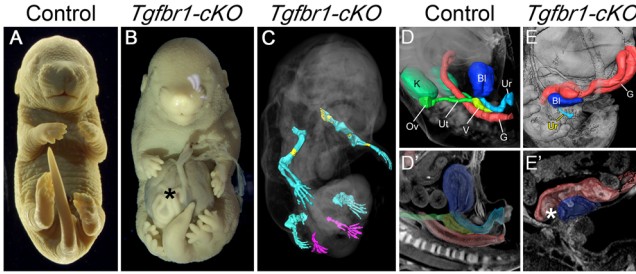

**Fig. 1 | Malformations in the E16.5 *Tgfbr1-cKO* fetuses.** Images of fixed control (**A**) and *Tgfbr1-cKO* (**B**) fetuses. The mutant fetus shows the presence of an omphalocele (asterisk) and hindlimb duplication. **C** 3D reconstruction of the limb skeleton of a *Tgfbr1-cKO* fetus obtained by OPT and after segmentation of the limb skeleton. Extra hindlimbs are in magenta. Ossification shown in yellow. **D–E'** 3D reconstruction of organs and the excretory outlets of the control (**D**, **D'**) and *Tgfbr1-cKO* (**E**, **E'**) fetuses. Images were obtained by OPT, followed by segmentation of the relevant structures. **D' E'** show virtual sections of the segmented 3D specimen. Asterisk in **E'** shows the gut-bladder connection in the mutant. K kidney, G gut, Bl bladder, Ov ovary, Ut uterus, V vagina, Ur urethra.

last two sections were analyzed in more detail using light sheet microscopy, in addition to the presence of two skeletal elements in the zeugopod, we could identify individual phalanges in several of the fingers, although with somewhat abnormal patterns, as well as polydactyly (Supplementary Fig. 2).

Another prominent phenotype of *Tgfbr1-cKO* fetuses was the presence of an omphalocele [5/5 at E16.5] (Fig. 1B). This phenotype, but not the duplicated hindlimbs, was also observed in *Tgfb2/Tgfb3* double mutants[54]. This indicates both that Tgfbr1 might mediate Tgfb2 and Tgfb3 activity in the body wall and that different ligands should trigger Tgfbr1 signaling in the body wall and the pericloacal/hindlimb mesoderm. The three *Tgfbr1-cKO* mutants analyzed either by optical projection tomography (OPT) or by histological sections lacked kidneys (Fig. 1D–E'; Supplementary Movies 2 and 3), a trait also observed in mutant fetuses for *Gdf11*[2,39]. The cloacal-derived tissues were also strongly compromised in these mutant embryos (Fig. 1D–E'; Supplementary Fig. 3C, C'; Supplementary Movies 2 and 3). The gut failed to generate normal rectal-anal structures. Instead, the caudal end of the intestinal tube merged with a structure that could be identified as a small bladder connected to a hypomorphic urethra, thus resembling a persistent cloaca. The external genitalia were also virtually absent from *Tgfbr1-cKO* fetuses. These phenotypes are likely to result from interference with the functional interactions between the cloacal endoderm and the pericloacal mesoderm driving normal morphogenesis of the cloacal region[9,11,18,19,31]. Indeed, endodermal malformations in this area were already evident in *Tgfbr1-cKO* mutants at mid gestation. At E10.5 and E11.5, the mutant cloaca was larger than in control embryos, lacked features associated with normal septation, and often contained protrusions entering the intra-cloacal lumen (2/4 embryos, Supplementary Fig. 3a–B'). Remains of these protrusions persisted later in development at E16.5 in the malformed bladder (Supplementary Fig. 3C, C').

**The extra set of hindlimbs in Tgfbr1-cKO mutants derives from the pericloacal mesoderm**

Morphological alterations in the hindlimb/pericloacal area of *Tgfbr1-cKO* embryos were already visible at mid-gestation (Fig. 2). *Fgf8* expression in E10.5 embryos revealed a posterior and medial expansion of the apical ectodermal ridge (AER) of the hindlimb bud almost reaching the ventral midline of the embryo (Fig. 2A, A'), suggesting an extension of the hindlimb field into the pericloacal mesoderm of *Tgfbr1-cKO* embryos. This was associated with several expression changes in the pericloacal region. For instance, hindlimb specific markers, like *Pitx1* and *Lin28a* were at E10.5 ectopically activated in the

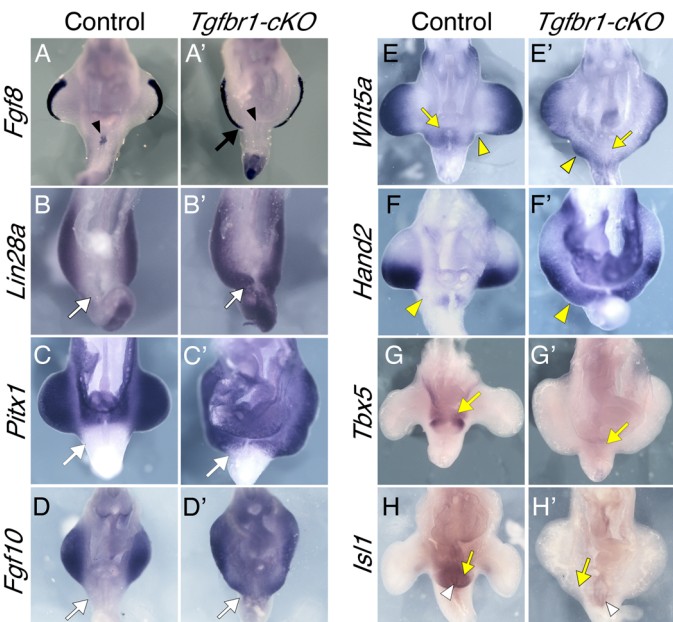

**Fig. 2 | The pericloacal mesenchyme of the *Tgfbr1-cKO* embryos adopts hindlimb fate.** Ventral views of the hindlimb/cloacal region of wild type (**A–H**) or *Tgfbr1-cKO* embryos (**A'–H'**) stained for *Fgf8* (E10.5)(**A**, **A'**), *Lin28a* (E10.5) (**B**, **B'**), *Pitx1* (E10.5) (*n* = 2); **C**, **C'**), *Fgf10* (E10.5) (**D**, **D'**), *Wnt5a* (E11.0) (**E**, **E'**), *Hand2* (E11.0) (**F**, **F'**), *Tbx5* (E11.5) (**G**, **G'**) or *Isl1* (E11.5) (**H**, **H'**). *Fgf8* expression shows posterior and medial extension of the AER (black arrow) and the absence of expression in the cloacal endoderm (black arrowhead). Hindlimb markers *Lin28a*, *Pitx1* and *Fgf10* extend into the pericloacal region *Tgfbr1-cKO* embryos (white arrows). *Wnt5a* and *Hand2* are expressed in the posterior prominence of the *Tgfbr1-cKO* hindlimbs, while forming clear separation of the hindlimb and GT domains in the control embryos (yellow arrowheads). Pericloacal expression of *Wnt5a*, *Tbx5* and *Isl1* is lost in the mutant embryos (yellow arrows). *Isl1* expression in cloacal endoderm is marked by white arrowhead. At least *n* = 3 embryos were analyzed per probe, unless stated otherwise, giving equivalent patterns.

pericloacal region (Fig. 2B–C'). Conversely, *Tbx5* expression, normally observed in the pericloacal mesoderm, and later in the GT, but not in the hindlimb, could not be detected in this region of E11.5 *Tgfbr1-cKO* embryos (Fig. 2, G'), and *Fgf8* expression was absent from the cloacal endoderm of the mutant embryos (Fig. 2A, A'). Together these observations suggest a change of identity in the pericloacal mesoderm, becoming incorporated into the hindlimb field instead of entering its normal GT developmental fate.

Expression of genes active in both the hindlimb and the pericloacal region/GT, was also consistent with the pericloacal tissues taking a hindlimb fate. For instance, *Wnt5a*, which in wild type embryos is expressed in the limb buds beneath the AER and in the pericloacal region and GT, was not detected next to the cloacal endoderm of E10.5 *Tgfbr1-cKO* embryos, being instead observed beneath the AER along the extended hindlimb domain into the pericloacal region (Fig. 2E, E'). This pattern also revealed that the hindlimb domain of *Tgfbr1-cKO* embryos become split in two prominences, the first indication of the generation of two individual hindlimb structures from extended hindlimb bud of the mutant embryos. *Fgf8* expression in this region of *Tgfbr1-cKO* mutants at E11.5 was also consistent with the split of the hindlimb bud in two fields, as it was observed through the apical border of the hindlimb region but demarcating two apparent prominences separated by a valley (Fig. 3A, A'). Additional support for the conversion of pericloacal mesoderm into hindlimb identity was provided by genes showing dynamic expression patterns in these structures. In wild type embryos, *Isl1* is initially activated in the hindlimb and GT primordia, becoming later downregulated in the hindlimb while keeping strong expression in the GT[55] (Fig. 2H). At E11.5 *Tgbfr1-cKO*

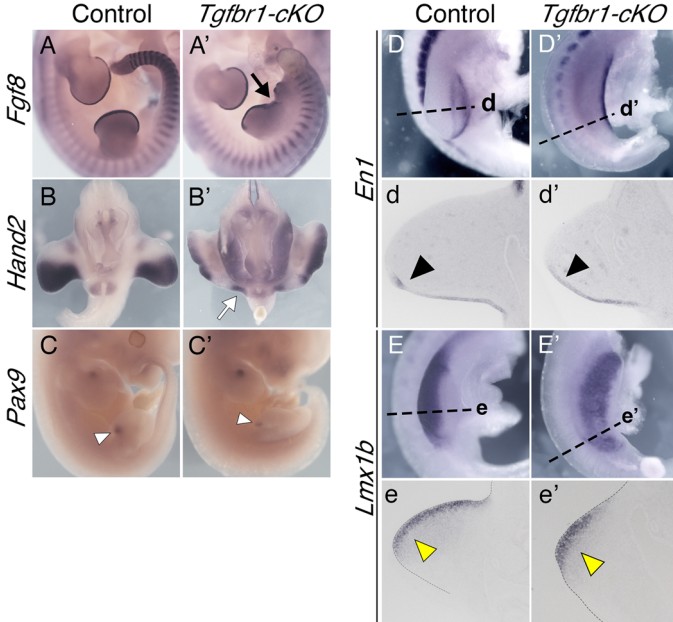

**Fig. 3 | Limb patterning in the *Tgfbr1-cKO* embryos.** Expression of *Fgf8* in E11.5 control (**A**) and *Tgfbr1-cKO* (**A′**) embryos. The black arrow shows separation between the two hindlimb prominences. Expression of *Hand2* in E11.5 control (**B**) and *Tgfbr1-cKO* (**B′**) embryos. White arrow in (**B′**) shows a posterior expression domain of the prospective mutant extra hindlimb. Anterior *Pax9* expression in E11.5 control (**C**) and *Tgfbr1-cKO* (**C′**) hindlimbs. A single expression signal is observed in the hindlimb region of *Tgfbr1-cKO* embryo corresponding to the most anterior part of the structure (white arrowhead). **D**–**e′** Dorsal-ventral polarity in the *Tgfbr1-cKO* mutant's hindlimb. Expression of the ventral limb marker *En1* in E10.5 control (**D**, **d**) and *Tgfbr1-cKO* (**D′**, **d′**) embryos. **D**, **D′** show lateral views of the hindlimb region in whole mount embryos; (**d**, **d′**) show transversal sections through the region indicated in **D**, **D′**. Black arrowheads show *En1* expression in the ventral ectoderm of the limb bud. Expression of the dorsal mesodermal limb marker *Lmx1b* in E10.5 control (**E**, **e**) and *Tgfbr1-cKO* (**E′**, **e′**) embryos. **E**, **E′** show lateral views of the hindlimb region in whole mounted embryos; (**e**, **e′**) show transversal sections through the region indicated in **E**, **E′**. Yellow arrowheads show *Lmx1b* expression in the dorsal mesenchyme of the limb bud. *n* = 3 embryos were analyzed per probe giving equivalent patterns.

mutants showed residual *Isl1* expression mainly in the expanded endodermal cloaca (note that *Isl1* normal expression profile includes the cloacal endoderm[35]) but became downregulated in the adjacent mesoderm in the area likely to generate the extra hindlimb bud (Fig. 2H′). Conversely, *Fgf10*, an essential gene for limb induction[56], whose expression in the GT is observed only at later stages[26], was strongly activated already at E10.5 following limb-like patterns, reaching mesodermal tissue adjacent to the endoderm (Fig. 2D, D′).

Although clearly not as well defined as in wild type embryos, a variable degree of anterior-posterior patterning could still be observed in the hindlimb region of *Tgfbr1-cKO* embryos, consistent with changes in digit identity observed in the autopod skeleton of the E16.5 fetuses (Supplementary Fig. 2). For instance, at E10.5, *Hand2* was expressed in the mutants as a continuous domain that run medially into pericloacal tissue, showing a slightly stronger expression in the posterior area of the two emerging prominences of the hindlimb bud, although not as clearly restricted to the posterior limb bud mesenchyme observed in wild type embryos (Fig. 2F, F′). A similar trend was observed at E11.5, when *Hand2* expression followed a slight anterior-posterior distribution, being excluded from the anterior-most region of both anterior and posterior limb bud prominences (Fig. 3B, B′) and showed a slightly stronger expression in the posterior part of the caudal prominence of the mutant limb bud (Fig. 3B, B′). Expression of the anterior marker *Pax9* was observed in the anterior border of the anterior prominence

of the limb bud in late E11.5 *Tgfbr1-cKO* embryos (Fig. 3C, C′). However, we could not detect *Pax9* signal associated with the posterior prominence. Whether this indicates real absence of *Pax9* expression in this region or just delayed onset of expression remains to be determined.

Interestingly, the posterior prominence of the mutant hindlimb bud retained a considerable degree of dorso-ventral patterning, as revealed by *En1* and *Lmx1b* expression at E10.5 that followed typical limb patterns[57,58], although they were not as well defined as in control hindlimb buds (Fig. 3D–e′).

It has been shown that regulation of GT development relies largely on signals from the cloacal endoderm, with Shh playing a central role in this process[30,31]. The observation that *Wnt5a*, a known Shh target in the GT[31], was not activated in the pericloacal mesoderm of *Tgfbr1-cKO* despite strong endodermal *Shh* expression (Fig. 4A, A′; Supplementary Fig. 3a–B′), suggested that either this mesenchyme became refractory to Shh activity or that it changed its response profile. The expression of the Shh target *Gli1* was consistent with the second possibility, as it showed equivalent patterns in the pericloacal mesenchyme in both wild type and *Tgfbr1-cKO* embryos (Fig. 4B, B′), thus indicating that this tissue still responds to the Shh signal from the endoderm.

### Reorganization of the chromatin regulatory landscape in the pericloacal mesoderm of Tgfbr1-cKO embryos

We explored the mechanisms modulating the pericloacal mesoderm response to Shh (and most likely also to other regulatory factors) by comparing the genomic accessibility profiles in the posterior hindlimb prominences (extra hindlimbs) and region adjacent to the cloaca (we will refer to it as mutant GT) of *Tgfbr1-cKO* with those of hindlimb buds and GT of control embryos by ATAC-seq[59] (Fig. 4C). Principal component (PC) analysis of those profiles revealed two main features. PC1 grouped the tissues from *Tgfbr1-cKO* embryos together with the wild type GT and clearly apart from the control hindlimb, consistent with the pericloacal origin of the extra hindlimb of the mutant embryos. PC2, on the contrary, grouped the extra hindlimb of the mutant embryos together the wild type hindlimb, separated from the control GT (Fig. 4D).

We further explored the PC1 and PC2 by annotating 500 top peaks contributing for each of the PCs (top loadings). For both PC1 and PC2, most of the top loadings were in distal intergenic regions (46.2% and 48.6%, respectively), while only small proportion of the peaks (6,8% for PC1 and 8,6% for PC2) were attributed to potential promoter regions (<=5 kb upstream of TSS) (Fig. 4E). This suggested that most of the genomic regions whose accessibility patterns are influenced by the Tgfbr1 might represent distal regulatory elements.

Hierarchical clustering of the top loadings contributing to the PC1 reveled the existence of two distinct clusters: cluster 1 represent peaks less accessible in the control limb than in other conditions, while cluster 2 contains the peaks more accessible in the control limb than in other conditions (Fig. 4F). These two clusters represent limb specific regions regulated by Tgfbr1 signaling. Top loadings of PC2 formed 4 clusters based of their accessibility patterns (Fig. 4G). Clusters 1 and 3 are particularly interesting, because they represent the peaks with accessibility patterns common between control and mutant limbs, and different from wild type and mutant GT. These clusters, therefore, are likely to contain the regulatory elements specifying limb fates and being responsible for directing the mutant pericloacal mesenchyme towards the limb fate.

To further elaborate on the potential significance of conserved patterns within top PC loadings clusters, we performed a series of complementary differential analyses on our samples. We identified two accessibility patterns suggestive of their involvement in the differential control of cell fate decisions of the pericloacal mesoderm. First, we identified a set of peaks more accessible in the GT than in the hindlimb samples from wild type embryos, and that showed significantly reduced accessibility in the tissues from *Tgfbr1-cKO* embryos

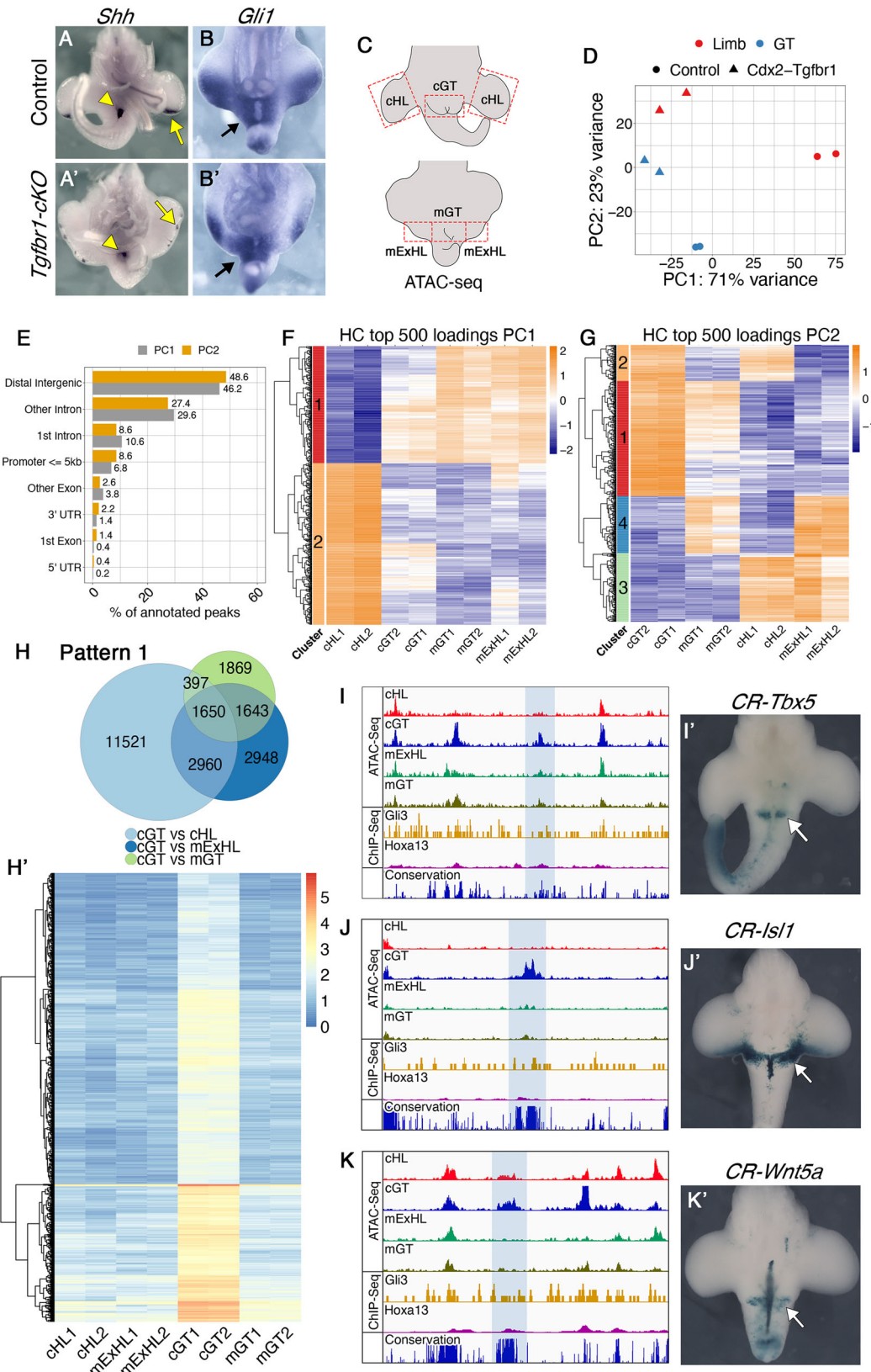

(1650 regions with $Log_2FC > 1.5$; FDR $< 1e^{-3}$) (Fig. 4H, H'). These elements might represent enhancers involved in the regulation of genes differentially expressed in the pericloacal region of wild type embryos. The loss of accessibility to these elements in the *Tgfbr1-cKO* mutants could thus contribute significantly to the inability of the pericloacal mesoderm of the mutant embryos to enter their normal GT fate.

Another group of chromatin regions were significantly more accessible in hindlimb-generating tissues of wild type and *Tgfbr1-cKO* embryos than in GT tissues (526 peaks with $Log_2FC > 1.5$; FDR $< 1e^{-3}$ are shown in Fig. 5A, A'). These regions represent a pattern analog to that observed in clusters 1 and 3 of PC2 whose accessibility patterns did not diverge between mutant and control hindlimb ($Log_2FC < 0.5$; FDR >

**Fig. 4 | Tgfbr1 regulates the chromatin landscape in the pericloacal region. A–B'**
The pericloacal mesenchyme of *Tgfbr1-cKO* embryos responds to Shh signaling
from the cloacal endoderm. *Shh* expression in E11.5 control (**A**) and *Tgfbr1-cKO* (**A'**)
embryos. *Gli1* expression in E10.5 control (**B**) and *Tgfbr1-cKO* (**B'**) embryos (*n* = 3
embryos were analyzed per probe). **C** Schematic representation of the tissue dis-
sected for the ATAC-seq experiments in control (top) and mutant (bottom)
embryos. **D** PCA plot of the ATAC-seq samples. Limb samples are shown in red; GT
samples are in blue; circle – control; triangle – *Tgfbr1-cKO*. **E** Frequencies at the
annotated genomic regions of 500 top loadings from the PC1 (gray) and PC2
(yellow). **F, G** Hierarchical clustering (HC) of PC1 (**F**) and PC2 (**G**) 500 top loadings.
Heatmaps show scaled normalized counts. **H, H'** Chromatin regions following
pattern 1. **H** Venn diagram showing interception of the regions tested more
accessible in control GT than in other tissues by pairwise comparisons [FDR <

0.001, logFC > 1.5]. **H'** Heatmap showing log2 normalized counts across samples of
the regions following pattern 1. **I–K** ATAC-seq tracks showing accessible chromatin
in control GT, but not in the other samples in a region 93.7 kb upstream of the *Tbx5*
TSS (**I**), 24.0 kb downstream of the *Isl1* TSS (**J**), and 91.7 kb upstream of the *Wnt5a*
TSS (**K**). The regions highlighted with the blue shade are highly conserved among
placental animals (bottom track). Two independent biological replicates were
analyzed per tissue in the ATAC-seq experiments. **I', J', K'** The highlighted elements
drive β-galactosidase expression in the GT region (white arrows). Hoxa13 and Gli3
ChIP-Seq tracks from forelimb buds (GSE81356 and GSE133710, respectively) are
shown in yellow and magenta, respectively. None of the analyzed enhancers
showed enrichment in Hoxa13 or Gli3 binding activity in the forelimbs, consistent
with the GT specificity of those elements. cHL control hindlimb, mExHL mutant
extra hindlimb1cGT control GT, mGT mutant GT, CR conserved region.

0.1), indicating that they might represent the most relevant elements
driving the mutant pericloacal mesenchyme towards limb identity. For
further description we will refer to the GT-specific elements as pattern
1 elements and to those associated with hindlimb tissues pattern 2
elements.

We identified pattern 1 elements within the *Tbx5, Isl1* and *Wnt5a*
genomic regions, three of the pericloacal mesoderm markers down-
regulated in the pericloacal region of *Tgfbr1-cKO* mutants (Fig. 4I, J, K),
which will serve as a proof of principle for the involvement of pattern 1
elements in GT expression. These elements mapped to highly con-
served genomic areas, suggesting their possible involvement in reg-
ulatory processes. When tested using a transgenic reporter assay, one
of the elements associated with *Tbx5* showed activity in the pericloacal
region [*n* = 6/14] (Fig. 4I'). Interestingly, it did not activate overt fore-
limb expression (Supplementary Fig. 4A), the major expression
domain of this gene[14], further indicating specificity for the pericloacal
region. Similarly, the element identified downstream of *Isl1* also largely
reproduced *Isl1* expression in the GT when tested in transgenic
reporter assays [*n* = 5/12] (Fig. 4J', Fig. 6B, and Supplementary Fig. 4C).
An element associated with *Wnt5a* was also able to activate expression
in the GT, although much less frequently than the other elements
[*n* = 2/18] (Fig. 4K'). This element was also active in the tail, but it did
not promote expression in the hindlimb. Together, these observations
are consistent with these elements indeed representing enhancers
potentially involved in the expression of the relevant genes in the
pericloacal area.

We found pattern 2 elements in regions associated with genes
playing essential roles during the earliest stages of limb development,
which could thus play a relevant role in promoting limb fates from the
pericloacal mesoderm (Fig. 5C, D). As for pattern 1 elements, these
regions were also highly conserved among vertebrates. We found
pattern 2 elements in the *Fgf10* genomic region (Fig. 5C), a gene acti-
vated in the mesenchyme adjacent to the cloaca entering hindlimb fate
in the mutant embryo (Fig. 2D, D'). Interestingly, published ChIP-seq
data from forelimb buds[60,61] showed binding of Gli3 and Hoxa13 to one
of these elements, located within an intronic region of *Fgf10* (Fig. 5C),
indicating that it might respond to Shh and/or Hoxa13 activities in the
developing limb buds. The lack of accessibility of this element in the
GT of wild type embryos might thus suggest that Tgfbr1 signaling
renders this enhancer blind to endodermal Shh and/or the strong
pericloacal Hoxa13 (and maybe also Hoxd13) expression[34], consistent
with absent *Fgf10* at early stages in this tissue (Fig. 2D). Rather sur-
prisingly, this element failed to generate reporter activity in transgenic
embryos [*n* = 10]. We still do not understand whether this reflects real
lack of activity or the absence of proper genomic context for their
activity.

Another potentially relevant pattern 2 element was located within
the *Fmn1* locus, matching the position of one of the enhancers (ele-
ment 7 in ref. 62) controlling *Grem1* in the limb buds (Fig. 5D) that had
been shown to respond to Shh activity from the zone of polarizing
activity (ZPA)[62,63]. In wild type embryos *Grem1* expression is not

detected in the pericloacal mesoderm or in the GT (Fig. 5E), indicating
that in these tissues *Tgfbr1* might render the *Grem1* regulatory ele-
ments blind to endodermal Shh. The inaccessible state of this *Grem1*
enhancer in the GT (Fig. 5D) is consistent with this hypothesis. Con-
versely, the gain of accessibility of the *Grem1* enhancer in the extra
hindlimb suggests that in the absence of *Tgfbr1* the pericloacal
mesoderm could become responsive to the endodermal Shh. Con-
sidering the capacity of *Grem1* to expand the expression of AER
markers[64–66], this gene is a candidate to contribute to the observed AER
extension in the hindlimb primordium of *Tgfbr1-cKO* embryos. *Grem1*
expression analyses in *Tgfbr1-cKO* embryos revealed unexpected
complexities. Some features of the expression pattern were consistent
with *Grem1* becoming responsive to endodermal Shh, as expression
was detected in the caudal-most end of the expanded hindlimb next to
the endoderm (Fig. 5E'). Rather surprisingly, however, *Grem1* expres-
sion was strongly reduced or absent throughout the mesoderm of the
extended hindlimb bud of *Tgfbr1-cKO* embryos (Fig. 5E'). *Grem1* limb
expression depends on Shh activity from the ZPA[62,67,68], suggesting that
the absent *Grem1* expression resulted from the abnormal *Shh* expres-
sion observed in the hindlimb buds of *Tgfbr1-cKO* mutants, which
followed spotty patterns beneath the AER instead of forming a well-
defined ZPA in the posterior limb bud mesenchyme (Fig. 4A', Supple-
mentary Fig. 3A').

We explored whether this abnormal *Shh* expression resulted from
a functional impact of the absence of Tgfbr1 activity on the ZRS, the
regulatory element controlling *Shh* expression in the ZPA[69]. As
expected, this region was accessible in wild type hindlimbs (Fig. 5F).
Conversely, it was inaccessible in the wild type GT, consistent with the
absence of the *Shh* expression in the pericloacal mesoderm or the GT
mesoderm, despite the presence of relevant activators of this enhan-
cer, like Hand2, Hoxa13 and Hoxd13[17,34,70,71] [the endodermal *Shh*
expression is under the control of a different enhancer[72], which was
accessible in both GT samples (Supplementary Fig. 3D), consistent
with conserved endodermal *Shh* expression in the *Tgfbr1-cKO*
embryos]. Remarkably, the ZRS was also non-accessible in the hin-
dlimb region of *Tgfbr1-cKO* embryos, consistent with the observed
abnormal *Shh* limb expression (Fig. 5F). Therefore, Tgfbr1 seems to
also control ZRS accessibility to their regulatory transcription factors.
Interestingly, this enhancer is part of an additional pattern (pattern 3)
consisting of elements requiring Tgfbr1 activity to become accessible
in the limb bud. Analysis of the ATAC-seq data revealed that 5764
(Log2FC > 1.5; FDR < 1e$^{-3}$) elements fit the criteria of this pattern
(Fig. 5B, B'). The complementary tissue specificities of patterns 1 and 3
indicates context dependency for Tgfbr1 activity in the modulation of
genomic configuration.

**Footprinting analysis on elements differentially accessible
in the GT**
The canonical Tgfbr1 signaling pathway involves activation of Smad2
or Smad3 that recruit Smad4 and enter the nucleus where they interact
with Smad-responsive elements to regulate gene expression[73]. It has

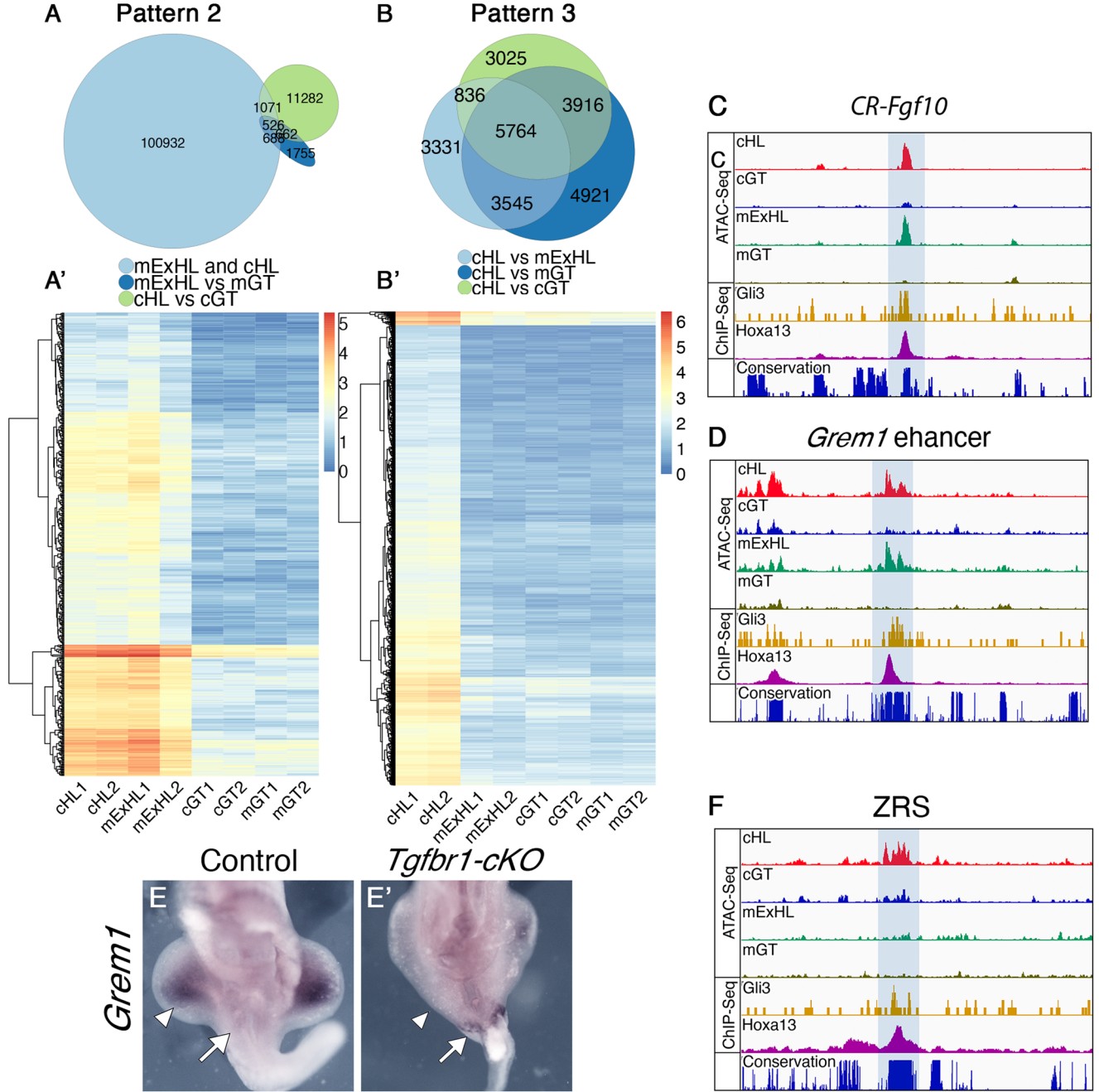

**Fig. 5 | Tgfbr1 regulates accessibility of limb regulatory regions. A**, **A′** Chromatin regions following pattern 2 and pattern 3. **A** Venn diagram showing interception between the regions more accessible in mutant extra limb than in mutant GT [FDR < 0.001, logFC > 1.5] and more accessible in control hindlimb than in control GT [FDR < 0.001, logFC > 1.5], but at the same time not differentially accessible between mutant extra hindlimb and control hindlimb [FDR > 0.01, logFC < 0.5]. **A′** Heatmap showing log2 normalized counts from genomic regions following pattern 2 across the samples. **B**, **B′** Chromatin regions following pattern 3. **B** Venn diagram showing interception between regions tested more accessible in control hindlimb than in other tissues by pairwise comparisons [FDR < 0.001, logFC > 1.5]. **B′** Heatmap showing log2 normalized counts for the chromatin regions following pattern 3 across samples. **C**, **D** ATAC-Seq profiles showing *Fgf10* putative regulatory element (**C**) and *Grem1* enhancer (**D**) that gains accessibility in the mutant extra hindlimb (highlighted with the blue shadow). Also shown are ChiP-Seq tracks of ChIP seq data

from wild type forelimbs for Gli3 (yellow) and Hoxa13 (magenta) (obtained from GSE81356 and GSE133710, respectively), indicating their binding to those elements in the forelimb bud. The lower track shows conservation in placental mammals. *Grem1* expression in E10.5 wild type (**E**) or *Tgfbr1-cKO* (**E′**) embryos showing the absence of pericloacal expression in wild type embryos and ectopic activation in the caudal-most region of the pericloacal mesoderm of the mutant embryo (white arrow). *Grem1* is expressed in the developing limb bud of the control embryo, but it is almost not detectable in the developing limb buds of the mutant embryo (white arrowheads) (*n* = 3 embryos are analyzed per probe). **F** ATAC-seq data through the ZRS (highlighted with the blue shadow), showing that it is accessible in the control limb but not in any of the other samples. Two independent biological replicates were analyzed for all ATAC-seq experiments. CR conserved region, cHL control hindlimb, mExHL mutant extra hindlimb, cGT control GT, mGT mutant GT.

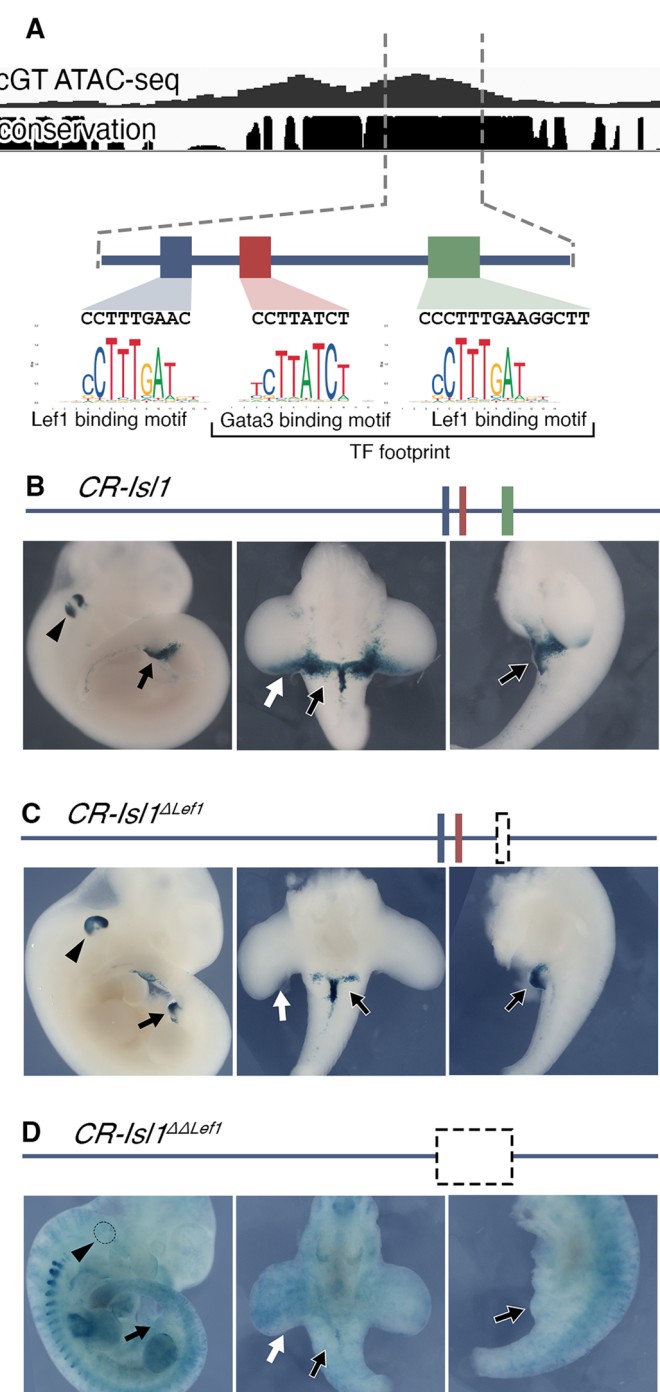

**Fig. 6 | *CR-Isl1* is a potential Wnt-responsive element. A** Schematic representation of the *CR-Isl1* region of Fig. 4J. The top panel shows the ATAC-seq signal in control GT and the placental conservation score. The footprint corresponded to a -65 bp region containing a Gata3 binding motif (red), and Lef1 binding motif (green). An additional Lef1 binding motif (blue) was identified 21 bp upstream of the Gata3 motif, although it was not included within the footprint. The genomic sequences of these motifs are shown along with position frequency matrices (PFMs). Activation of the β-galactosidase reporter by the *CR-Isl1* (**B**), *CR-Isl1*^DLef1 (**C**), and *CR-Isl1*^DDLef1 (**D**) elements in E11.5 embryos. Black arrowheads show expression in otic vesicle; white arrow indicates expression in the posterior limb bud; black arrow shows expression in the GT.

been shown that Smads can interact with chromatin remodelers, like Foxh1 or components of the SWI/SNF complex to promote accessibility of enhancers that are later regulated by Smad transcriptional activity[74–76]. To understand whether a similar mechanism might be

involved in the reorganization of the regulatory landscape in the pericloacal mesoderm we performed a transcription factor footprint analysis and subsequently motif enrichment on the ATAC-seq profiles using HINT-ATAC[77]. We did not find enrichment of Smad footprints in the accessible regions of the pericloacal mesoderm of wild type embryos (Supplementary Data 1). Similarly, the transcription factor Foxh1, previously shown to interact with Smad proteins was also not found enriched in these elements. These results indicate that the mechanism by which Tgfbr1 signaling modulates accessibility of regulatory regions is likely to be different from that described for the regulation by Tgfβ or Nodal in cell lines[74–76]. The lack of enrichment for Smad footprints in the elements specifically accessible in the GT of wild type embryos indicate that Smad-dependent signaling is unlikely to be a major component participating in the promotion of genital fates in the pericloacal mesoderm.

Additional analysis of these footprints revealed enrichment in signals associated with factors known to play significant roles in the development of the external genitalia, including Hoxa13, Hoxd13, Isl1, Six2 or effectors of the canonical Wnt pathway like Lef1, Tcf7 or Tcf7l. As a proof of principle to understand the relevance of these footprints in the regulation of pattern 1 elements, we focused on a Lef1 binding motif identified in the TF footprint analysis within the *CR-Isl1* chromatin region analyzed above (green rectangle in Fig. 6A). We tested the functional relevance of this potential Lef1 binding site, by generating a mutant version of this enhancer lacking this site (*CR-Isl1*^ΔLef1) and testing its activity in the β-galactosidase transgenic reporter assay (Fig. 6C and Supplementary Fig. 4D). When compared to the wild type enhancer (Fig. 6B, Fig. 4J' and Supplementary Fig. 4C) *CR-Isl1*^ΔLef1 transgenic embryos showed reduced activity in the GT mesenchyme, and no activity in the posterior part of the hindlimb typically observed in the control embryos [6/15] (Fig. 6C and Supplementary Fig. 4D). In addition, *CR-Isl1*^ΔLef1 activated β-galactosidase expression in the otic vesicle (Fig. 6C and Supplementary Fig. 4D), likely derived from a conserved Gata3 motif within the TF footprint (red rectangle in Fig. 6A) at level comparable to that observed in *CR-Isl1* transgenics (Fig. 6B, Supplementary Fig. 4C), which served as an internal control to evaluate reporter expression. This is consistent with potential relevance of this Lef1 binding site for the activity of this *Isl1* associated regulatory element, and thus, to respond to canonical Wnt signaling in the GT. It is possible that the residual activity of *CR-Isl1*^ΔLef1 in the GT derived from the presence of an additional Lef1 binding motif slightly upstream on the TF footprint (blue rectangle in Fig. 6A). We therefore tested the activity of a deletion mutant of this enhancer including both Lef1 binding motifs and the region between them (*CR-Isl1*^ΔΔLef1). Transgenic embryos for this construct had no sign of activity in the GT, limbs, or the otic vesicle [2/7], promoting instead extensive reporter activation throughout the embryo or in different parts of the embryo unrelated to the areas of activity associated with the *CR-Isl1* enhancer (Fig. 6D, Supplementary Fig. 4E). This pattern contrasts with those obtained with *CR-Isl1* or *CR-Isl1*^ΔLef1, which were consistently almost restricted to the GT and otic vesicle. Together, these results are compatible with the interpretation that Tgfbr1 signaling renders the *CR-Isl1* enhancer accessible to its regulation by canonical Wnt signaling in the pericloacal region.

## Discussion

The capacity of the pericloacal mesoderm of mouse embryos to generate both external genitalia and hindlimb structures reveals a remarkable degree of developmental plasticity in this tissue. This finding provides an additional dimension to the developmental connection between the hindlimb and the GT, previously suggested by the considerable overlap in their regulatory networks[9,17–19]. It should be noted that *Tgfbr1-cKO* embryos develop two independent hindlimb structures despite being first induced seemingly as a single bud with a common AER extending into the pericloacal mesoderm. The PC analysis of ATAC-seq profiles showed that the tissue eventually generating

the extra hindlimb keeps a high degree of similarity with the GT of wild type embryos (PC1 in our analysis), despite the simultaneous loss of some GT signatures and the acquisition of several limb-like characteristics (PC2 in our analysis). It is thus possible that the tissues originating from the pericloacal mesoderm and the somatopleure-derived limb field contain distinct cell affinity properties that would prevent the two tissues from merging into a single hindlimb bud, thus generating two independent structures when exposed to limb-promoting inputs. Alternatively, the two independent limb structures could result from a discontinuous patterning process already present during the earliest stage of limb primordia. Indeed, the expression profiles of patterning genes show discontinuities between the anterior and posterior prominences of the mutant hindlimb field prior to limb morphogenesis. Further experiments will be required to properly understand the cellular and molecular basis of the formation of two instead of a single larger hindlimb structure in *Tgfbr1-cKO* fetuses.

The Tgfbr1-dependent differentiation plasticity of the pericloacal mesoderm, together with its apparent inability to mix with the primordial limb mesoderm has important implications for both the development and evolution of the limb and genital area of tetrapods. It has been suggested that in mice the posterior displacement of the developing cloaca relative to the hindlimb was key to disconnect the hindlimb field from the influence of the cloacal signals that induce the GT, thus allowing the somatic lateral plate mesoderm and the ventral lateral mesoderm entering hindlimb and genital fates, respectively[20]. This contrasts with the position of those structures in squamates, in which the cloaca is formed just adjacent to the hindlimb field, a configuration that has been suggested to promote development of paired external genitalia (hemipenises) instead of hindlimbs from these primordia in snakes[20]. However, similar proximity was also observed in lizards that develop both paired external genitalia and hindlimbs[20]. In addition, and most importantly, the phenotype of the *Tgfbr1-cKO* embryos shows that the mesoderm adjacent to the cloaca can form limb structures, indicating that the intrinsic properties of the mesoderm are key determinants of the differentiation route it will enter. It will be therefore interesting to determine whether a mechanism related to the developmental plasticity uncovered by our work, including possible changes in cell affinity properties of prospective hindlimb and genital primordia, could help explaining the absence of hindlimbs in snakes but their presence in most lizards.

We showed that *Tgfbr1* modifies the profile of chromatin accessibility in the pericloacal mesoderm. The involvement of Tgfβ/BMP signaling in chromatin remodeling has been reported before for Tgfβ and Nodal signals[74–76]. In those cases, this resulted from the interaction of Smads with chromatin remodelers like Foxh1 or components of the SWI/SNF complex, which would turn Smad-responsive enhancers accessible and enable the Smad transcriptional activity[74–76]. Although a similar mechanism could contribute to Tgfbr1 activity in the pericloacal and limb mesoderm, so far, we did not find evidence showing involvement of Smads in this developmental context. We did not detect enrichment of Smad binding to elements rendered accessible by *Tgfbr1* using HINT-ATAC[77] on the ATAC-seq profiles. Instead, this analysis showed some enrichment in footprints for factors known to play important roles in GT development. Even considering the limitations of the bioinformatic approach, these observations indicate that Tgfbr1 establishes the collection of regions accessible and inaccessible to regulatory factors, which are clearly not restricted to the downstream effectors of the Tgfbr1 pathway. It should also be considered that many chromatin regions gain accessibility in the absence of *Tgfbr1*, indicating that Tgfbr1 activity can also render chromatin regions non-accessible. However, the involvement of a Smad-dependent process in the layout of the chromatin accessible landscape of the pericloacal mesoderm cannot be ruled out, as Smad activity could be required at earlier developmental stages, controlling

the initial steps that establish the regulatory landscape without leaving recognizable signatures at later stages, and thus not present in our analyses.

Our work thus suggests a model according to which *Tgfbr1* controls the differential morphogenesis of the limb and GT primordia by determining the set of enhancers accessible or inaccessible to mediate the activity of patterning inputs. This regulatory mode gains relevance when considering that many of these inputs are common to both structures, despite generating mature structures as anatomically different as the hindlimbs and external genital structures[9,17–19]. This idea can be illustrated by two examples. One of them concerns the *Grem1* enhancer, involved in *Grem1* activation in the limb bud in response to Shh from the ZPA[62]. This enhancer is non accessible in the pericloacal mesoderm of wild type embryos, fitting with the lack of Grem1 expression in the pericloacal mesoderm of wild type embryos despite the strong Shh expression in the endodermal cloaca. It however became accessible in the *Tgfbr1*-negative mesoderm generating extra hindlimb, and *Grem1* expression was detected adjacent to the Shh source in the endoderm. The second example refers to the *Isl1* GT enhancer identified in this work. While we found no evidence of direct involvement of effectors of the Tgfbr1 pathway in its activation, our data indicate that *Tgfbr1* renders this enhancer accessible to regulation by the canonical Wnt signaling, a pathway that has been shown to participate in GT development[11].

The identification of chromatin elements uniquely associated with hindlimb and GT fates might guide the search for the molecular signature that specifies hindlimb or GT development. The large number of elements in each group suggest that these signatures might be quite complex, likely involving the combined activity of several factors. Identification of those factors will be challenging, considering that in mammals enhancers are typically located far from the genes they regulate, which are often not even the closest transcription unit[78].

An intriguing observation from our experiments was that *Tgfbr1-cKO* embryos were still able to generate mature limb structures that, although malformed, still contained most limb elements despite *Shh* being downregulated in the hindlimb bud. This phenotype contrasts with the absence of most distal elements in *Shh* mutant embryos[79]. It has been shown that a brief pulse of Shh activity during the initial bud stage is enough to promote normal limb development[63]. As the ZRS seems to be silent in *Tgfbr1-cKO* mutants, it is possible that the initial Shh input required to set limb development in motion is provided by the endoderm, facilitated by the extension of the hindlimb field into the pericloacal mesoderm. This would argue in favor of limb polarizing activity intrinsic to cloacal endoderm[31], and against exclusively pro-genital regulatory potential of cloaca[20].

A challenging question left open by our work is deciphering the mechanism(s) by which Tgfbr1 activity controls such large-scale remodeling of the regulatory landscape in the target tissues. The lack of enrichment in footprints for Tgfbr1 effectors in the target regions makes a direct activity of the canonical Tgfbr1 pathway in this process unlikely. Instead, it is more probable that this regulation is indirect. Whatever the mechanism or mechanisms, it requires a high degree of coordination, which would still be more challenging if several mechanisms are involved in the generation of the different patterns (e.g., if the control of accessibility and inaccessibility relies on independent mechanisms). Identification of those mechanisms and determining whether they also operate in other physiological and pathological processes under the control of members of the Tgfβ/BMP signaling family might have far reaching implications for our understanding of morphogenetic processes and disease.

## Methods

### Ethics statement
All animal procedures were performed in accordance with Portuguese (Portaria 1005/92) and European (directive 2010/63/EU) legislations

and guidance on animal use in bioscience research. The project was reviewed and approved by the Ethics Committee of "Instituto Gulbenkian de Ciência" and by the Portuguese National Entity "Direcção Geral de Alimentação Veterinária" (license reference: 014308).

## Mice

The mice used in this work were maintained on a 12 h light/dark cycle, at 22 °C with humidity ranging between 40 and 60%. Mice used to obtain embryos were between 3 and 6 months old. Generation of the *Tgfbr1*<sup>*Flox*</sup> mouse lines was previously reported[52]. *Tgfbr1*<sup>*3ex3-flox*</sup> resulted from the same experiments in which *Tgfbr1*<sup>*flox*</sup> was generated, identified upon sequencing through the *Tgfbr1* locus, which revealed that the exon 3 had been triplicated and the presence of LoxP sites in the introns between the triplicated exons and in those connecting them with exon 2 and exon 4. The *Cdx2Cre*<sup>*ERT*</sup> mice were previously reported[2]. All these mice had a mixed FBV/N-C57BL/6 background. Mice were genotyped from ear or digit biopsies. Samples were incubated in 50 μL of the PBND buffer (50 mM KCl, 10 mM Tris-HCl, pH 8.3, 2.5 mM MgCl₂, 0.1 mg/mL gelatin, 0.45% NP40, 0.45% Tween 20) supplemented with 100 μg/mL of proteinase K at 55 °C overnight. Proteinase K was inactivated at 95 °C for 15 min and 1 μL of genomic DNA was used for genotyping by PCR. *Cdx2Cre*<sup>*ERT*</sup>;*Tgfbr1*<sup>*+/−*</sup> mice were genotyped for the *Cre* allele and *Tgfbr1* null allele; *Tgfbr1*<sup>*flox/+*</sup> and *Tgfbr1*<sup>*3ex3-flox/+*</sup> mice were genotyped with primers surrounding 3'-LoxP (primers listed in Supplementary Table 1).

*Tgfbr1-cKO* (*Cdx2Cre*<sup>*ERT*</sup>;*Tgfbr1*<sup>*flox/−*</sup> or *Cdx2Cre*<sup>*ERT*</sup>;*Tgfbr1*<sup>*3ex3-flox/−*</sup>) embryos were obtained by crossing *Tgfbr1*<sup>*flox/flox*</sup> or *Tgfbr1*<sup>*3ex3-flox/+*</sup> females with *Tgfbr1*<sup>*+/−*</sup>;*Cdx2Cre*<sup>*ERT*</sup> males. Noon of the day of the plug was considered E0.5. To induce recombination pregnant females were treated with tamoxifen (Sigma, T5648) dissolved in corn oil. Tamoxifen was administered at 0.1 mg per gram of body weight by oral gavage either once at E6.75 when *Tgfbr1*<sup>*3ex3-flox*</sup> females were involved, or twice, at E6.75 and E7.25 when *Tgfbr1*<sup>*flox*</sup> females were involved.

Embryos were obtained by cesarean section and processed for further analyses (see below). They were genotyped from their yolk sacs for Cre, *Tgfbr1*<sup>*−*</sup> and *Tgfbr1*<sup>*flox*</sup> when analyzing *Tgfbr1-cKO* embryos or for β-galactosidase when analyzing the reporter transgenic embryos (primers listed in Supplementary Table 1). Specific analysis of the sex of the embryo was not included because at the stages analyzed the external genitalia of males and females have no sex differences. Yolk sacs were collected in 50 μL of yolk sac lysis buffer (50 mM KCl, 10 mM Tris-HCl, pH8.3, 2 mM MgCl₂, 0.45% Tween-20, 0.45% NP40) supplemented with 100 μg/mL of proteinase K and incubated at 55 °C overnight. Samples were heat-deactivated as previously described and used for PCR.

## Transgenic embryos

To generate transgenic constructs, the relevant enhancers (genomic coordinates shown in Supplementary Table 2) were amplified by PCR (primers in Supplementary Table 1) from genomic DNA and cloned into a vector containing the adenovirus major late promoter, the coding region of the β-galactosidase gene and the SV40 polyadenylation signal[2]. The enhancers were confirmed by direct sequencing. The constructs were isolated from the plasmid backbone, gel-purified using the NZYGelpure (NZYTech #MB01102) and eluted from the columns with 50 μL of the kit's elution buffer. The purified constructs were diluted in microinjection buffer (10 mM Tris.HCl, 0.25 mM EDTA, pH 7.5) at 2 ng/μL and microinjected into the pronucleus of fertilized FVB/N oocytes according to standard protocols[80]. Microinjected oocytes were transferred into the uteri of pseudopregnant NMRI females and embryos were recovered at E10.5, E11.5 or E12.0 and stained for β-galactosidase activity.

## RT-qPCR

To assess recombination efficiency, the posterior part of the E10.5 *Tgfbr1-cKO* embryos, including their hindlimbs, were dissected in PBS (137 mM NaCl, 2.7 mM KCl, 10 mM Na₂HPO₄, and 1.8 mM KH₂PO₄) on ice. Whole E10.5 *Tgfbr1*<sup>*−/−*</sup> embryos were used as controls. RNA was extracted from fresh tissue using TRI reagent (Sigma, #T9424) according to manufacturer's protocol. 200 ng of RNA from each sample was used to prepare complementary DNA using a random hexamer mix (NZYTech #MB12901) and following the protocol of the NZY Reverse Transcriptase enzyme (NZYTech #MB12401). All cDNA samples were diluted 1:5 and 1 μL was used in quantitative PCR (qPCR) reactions with iTaq Universal SYBR Green Supermix (Bio-Rad, #1725124), according to manufacturer's instruction and using the Applied Biosystems QuantStudio 7 flex qPCR System. Primers were localized in exon 3 and exon 4 of *Tgfbr1* mRNA (listed in Supplementary Table 1), so that in case of successful recombination the product is not amplified.

PCR product concentration in each sample was calculated using the standard curve with following formula $Quantity = 10^{(mean\ Ct-b/m)}$ (where "m" is slope and "b" is an intercept of the standard curve), and *Tgfbr1* expression level was normalized to *β-Actin* expression. Linear model was fitted (Tgfbr1 normalized expression ~Genotype) with R built-in function (lm). Analysis of variance (ANOVA) was performed on the linear model with the built-in function in R (anova). Significance levels between groups of samples were assessed by Tukey method using the "glht" function from "multicomp" package in R. Differences were considered significant at *$p < 0.05$, **$p < 0.01$ and ***$p < 0.001$. Boxplots were generated with "ggplot2" and "ggdignif" packages in R.

## Cell culture and transfection

HEK 293T cells (ATCC #CRL-3216) were cultured in Dulbecco's modified Eagle's medium (DMEM, Life Technologies) supplemented with 10% fetal bovine serum (FBS, Life Technologies) and 1% penicillin/streptomycin at 37 °C in a 5% CO₂ atmosphere. Cells were seeded in 35 mm plates and when they reached 70% confluency, the medium was changed to transfection medium (DMEM supplemented with 10% FBS). Expression vectors were constructed by amplifying the coding region of the *Tgfbr1* transcript isolated from wild type or *Tgfbr1*<sup>*3ex3-flox*</sup> homozygous embryos by RT-PCR (primers in Supplementary Table 1) and cloned into the pRK5 expression vector. Constructs were verified by sequencing. 3 μg of the plasmids was transfected into the 393T cells using Lipofectamine 2000 (Thermo Fisher Scientific #11668027) according to the manufacturer's instructions. After 24 h incubation, cells were processed for Western blot analysis.

## Protein extraction and western blot

The transfected cells were washed twice with ice-cold PBS and then 150 μL of lysis buffer (50 mM Tris-HCl, 150 mM NaCl and 1% Triton X-100, pH8.0) was added to each plate and incubated on ice for 10 min Lysates were scraped into microcentrifuge tubes on ice and centrifuged at 20000 rcf for 10 min at 4 °C. The supernatant was collected, frozen on dry ice and stored at −80 °C. Protein lysates were mixed (2:1) with 3X loading buffer (549 mM Tris HCl, 563 mM Tris Base, 2.05 mM EDTA, 8% LDS (Lithium dodecyl sulfate), 10% Glycerol, 0.75% SERVA Blue G250, 0.25% Phenol red, 160 mM DTT). Samples were incubated for 15 min at 65 °C and resolved by SDS-PAGE in a 10% polyacrylamide gel. Proteins were transferred to PVDF membranes in 20% methanol, 25 mM Tris, 200 mM glycine at 200 mA for 1 h. The membranes were blocked in blocking solution [5% dry milk dissolved in PBS containing 0.1% Tween-20 (PBT)] for 1 h at room temperature and then incubated with primary antibodies overnight at 4 °C. Primary antibodies were anti-Tgfbr1 (Sigma-Aldrich #HPA056473, 1:1000 in blocking solution) and anti-actin (Abcam #ab179467, 1:1000 in blocking solution). Membranes were then washed in PBT and incubated with HRP conjugated-anti-rabbit IgG (GE Healthcare # NA9340, 1:5000 in blocking solution) at room temperature for 1 h and washed twice in PBT. Signals were developed by chemiluminescence using Amersham ECL Prime Western Blotting Detection Reagent (GE Healthcare

#RPN2232) and images were captured using the GE Amersham Imager 600.

## Skeletal preparation

E17.5 fetuses were recovered from pregnant females by cesarean section and dissected from the extraembryonic membranes, eviscerated, the skin removed and then fixed in 100% ethanol for 2 days. Cartilages are then stained by incubation with 450 mg/L of alcian blue (Sigma, #A5268) in 80% ethanol/20% acetic acid for one day. Fetuses were then postfixed in 100% ethanol overnight. Tissue was cleared by incubation in 2% KOH for 6 h, followed by staining of ossified bones with a 50 mg/l of alizarin red S (Sigma, #130-22-3) solution in 2% KOH for 3 h. Specimens were then further incubated in 2% KOH until tissues were fully cleared. Skeletons were stored in 25% glycerol in water. All incubations were performed at room temperature with rolling. Genotyping was performed on gut tissue by incubating in Laird's buffer (100 mM Tris-HCl pH 8.5, 5 mM EDTA, 0.2% SDS, 200 mM NaCl) containing 10 µg/mL proteinase K at 55 °C overnight. Genomic DNA was then recovered by precipitation with isopropanol (1:1, vol:vol) and transferred to TE pH 8.0. Genotyping was then performed by PCR using oligos specified in Supplementary Table 1.

## OPT and SPIM imaging

Optical projection tomography (OPT) and SPIM (Selective Illumination Plane Microscopy) were used to image E16.5 fetuses as previously described[81] with minor modifications. Briefly, fetuses were recovered by cesarean section in ice cold PBS, washed several times in PBS and fixed in 4% paraformaldehyde (PFA) in PBS at 4 °C for several days. After several washes with demineralized water fetuses were dehydrated by sequential incubation in demineralized water with increasing concentrations of methanol (10% increases) and then twice in 100% methanol. Fetuses were then bleached in a three-day process with a sequence of 2.5%, 5% and 10% hydrogen peroxide in methanol at room temperature. Fetuses were then rehydrated through a reverse methanol/demineralized water series and embedded in a 0.7% low-melting agarose. Clearing was done with a 1:2 solution of Benzoic Alcohol:Benzyl Benzoate (BABB) using a BABB/methanol series (25% BABB increases). 100% BABB was introduced on day 3 and replaced every day for the next 4 days, until the fetuses became completely transparent. Anatomical datasets were then obtained using a custom built OPT/SPIM scanner and procedures; briefly, for OPT green auto-fluorescence was acquired on 1600 sequential angles for a full revolution, the raw dataset was pre-processed with a custom-built ImageJ macro, and then back-projection reconstructed using SkyScan's "nrecon" as in ref. 82, and then post-processed to reduce noise and enhance contrast in ImageJ. 3D reconstruction and visualization of internal organs was performed using the Amira software (Thermo Fisher Scientific). The autopods were also imaged with SPIM at higher magnification to be able to reconstruct the digits and bones, also using Amira. To facilitate the segmentation of the bones we used the Biomedisa online tool for machine-learning-assisted interpolation after manual segmentation of a few sparse sections[83]. The correlated OPT and SPIM datasets, and their segmented volumes, were then assembled into a correlative dataset in Amira, to produce the final 3D reconstructions.

## Histological analysis

E16.5 fetuses were fixed in Bouin's fixative at room temperature for 2 days, dehydrated by repeat washes in 100% ethanol for 5 days and embedded in paraffin. Paraffin blocks were then sectioned at 10 µm and stained with Masson's trichrome according to standard methods.

## Whole mount in situ hybridization and sectioning

Embryos were fixed in 4% PFA overnight, then dehydrated through 25%, 50% and 75% series of methanol in PBT (PBS, 0,1% Tween 20), then incubated in 100% methanol. Embryos were then rehydrated through a reverse methanol/PBT series and incubated three times in PBT for at least 5 min each at room temperature. Embryos were then bleached for 1 h in 6% hydrogen peroxide in PBT and permeabilized in 10 µg/mL of proteinase K in PBT for a period that depended on the embryo size. The reaction was then quenched with a 2 mg/mL solution of glycine in PBT, washed twice in PBT and postfixed in a 4% PFA and 0,2% glutaraldehyde mix for 20 min, followed by two washes in PBT. Hybridization was performed at 65 °C overnight in hybridization solution (50% formamide, 1.3 x SSC pH 5.5 [20 x SSC is 3M NaCl, 300 mM sodium citrate], 5 mM EDTA, 0.2% Tween 20, 50 µg/mL yeast tRNA, 100 µg/mL heparin) containing the relevant digoxigenin-labeled antisense RNA probes. The probes used in this work are listed in Supplementary Table 1. RNA probes were transcribed in vitro from the vector for 1.5 h at 37 °C with relevant RNA polymerase and the DIG RNA Labeling Mix (Roche #11277073910). The reaction product was verified in 0,8% agarose gel, precipitated with ethanol in the presence of 0.3M sodium acetate, pH 5.3 and resuspended in 100 µL of TE pH 8.0. Probes were diluted in hybridization solution (6 µL in 1 ml of solution). After hybridization, embryos were washed twice with hybridization solution without tRNA and heparin, at 65 °C for 30 min each, then in a 1:1 mix of hybridization solution and TBST (25 mM Tris.HCl, pH 8.0, 140 mM NaCl, 2.7 mM KCl, 0.1% Tween 20) for 30 min at 65 °C, and finally washed three times with TBST at room temperature. Embryos were then equilibrated in MABT (100 mM Maleic acid, 150 mM NaCl, 0.1% Tween-20, pH 7.5) and blocked in MABT blocking buffer [MABT containing 1% blocking reagent (Roche #11096176001)] with 10% sheep serum for 2.5 h at room temperature. Embryos were then incubated overnight at 4 °C with a 1:2000 dilution of alkaline phosphatase-conjugated anti-digoxigenin antibody (Roche #11093274910) in MABT blocking buffer with 1% sheep serum. After extensive washes with MABT at room temperature, embryos were equilibrated in NTMT buffer (100 mM Tris HCl, pH 9.5, 50 mM MgCl$_2$, 100 mM NaCl, 0.1% Tween-20) and developed with a 1:50 dilution of NBT/BCIP solution (Roche #11681451001) in NTMT at room temperature in the dark. The reaction was stopped with PBT, and embryos postfixed with 4% PFA overnight at room temperature and stored in PBT. Pictures were taken with a SWIFTCAM SC1803 connected to a stereoscope. Stained embryos were embedded in a mix containing 0.45% gelatin, 27% bovine serum albumin (BSA), 18% sucrose, jellified with 1.75% glutaraldehyde and sectioned at 35 µm on a Leica Vibratome VT 1000 S. Three biological replicates were analyzed per probe. Expression patterns were consistent among replicates, showing only small differences associated with the slightly variable shape of the hindlimb bud of mutant embryos.

## β-galactosidase staining

Embryos were dissected out in ice cold PBS and fixed in 4% PFA at 4 °C for 30 min They were then washed three times in β-gal wash buffer (PBS plus 0.02% Tween 20) for 10 min at room temperature and β-galactosidase activity developed in PBS containing 5 mM K$_3$Fe(CN)$_6$, 5 mM K$_4$Fe(CN)$_6$, 2 mM MgCl$_2$, 0.02% Tween 20, 0.4 mg/ml X-gal (Promega #V3941) overnight in the dark at 37 °C. The reaction was stopped with β-gal wash buffer, embryos postfixed in 4% PFA overnight at room temperature and stored in PBS.

## ATAC-Seq

ATAC-seq was performed as in ref. 59 with minor modifications. E11.25 mouse embryos were dissected on ice in DMEM High Glucose medium (Biowest #L0102-500) containing 10% FBS (this will be referred to as medium in the rest of the protocol). Hindlimbs and genital tubercles were dissected out and collected on ice-cold media. Single cell suspension was prepared by treating the tissue with 500 µL of Accutase (Sigma #A6964-500ML) for 30 min at 37 °C with shaking at 600 rpm. Single cells were pelleted at 6000 rcf for 5 min at 4 °C, resuspended in 200 µL of media and counted. 50000 viable cells from each sample

were used for nuclei extraction. Cells were incubated with 50 µL of ATAC resuspension buffer (10 mM Tris HCl pH 7.4, 10 mM NaCl, 3 mM MgCl$_2$, 0.1% NP-40, 0.1% Tween-20 (Sigma-Aldrich, P7949), 0.01% Digitonin (Target Mol #282T2721-1ml/10mM in DMSO) for 3 min on ice. Lysis was quenched by adding 1 ml of ATAC resuspension buffer without NP-40 and Digitonin. Nuclei were pelleted by centrifugation at 500 rcf for 10 min at 4 °C. The pellet was then resuspended in 50 µL of transposition buffer [for 50 µL: 25 µL 2x TD buffer, 2.5 µL Tn5 transposition enzyme (100 nM final) (Illumina #15028212), 16.5 µL PBS, 0.5 µL 1% digitonin, 0.5 µL 10% Tween-20, 5 µL H$_2$O] and incubated at 37 °C for 30 min with shaking at 1000 rpm. The DNA was purified with Qiagen MinElute PCR purification kit (Qiagen #50928004) and eluted in 20 µL of the kit's elution buffer. Libraries were amplified by PCR with NEBNext High-Fidelity 2X PCR Master Mix (New England Biolabs #174M0541S) for 9 cycles in 96-well Thermal cycler (Biorad) and purified with Qiagen MinElute PCR purification kit. Tagmentation efficiency was assessed on TapeStation 4200 (Agilent). Double size selection to remove primer dimers and fragments exceeding 1 kb was performed using SPRIselect beads (Beckman Coulter). Another quality control was performed with High Sensitivity DNA assay using the Fragment Analyzer (Agilent). The 4nM libraries pool was sequenced with Illumina NextSeq 2000 (100 cycles, Pair-end 50 bp). Two biological replicates were performed per tissue.

## ATAC-Seq bioinformatic and statistical analysis

Data bioinformatic analysis was performed on the Galaxy server[84]. Raw sequencing fastq files for each library were assessed for quality, adapter content and duplication rates with FastQC (v0.11.9) (http://www.bioinformatics.babraham.ac.uk/projects/fastqc). Adapters were trimmed and reads with length <20 bp were removed using cutadapt (v1.16.5)[85] (3′ adapter sequence: CTGTCTCTTATACACATCT). The trimmed reads were aligned to the mouse reference genome (GRCm38/mm10 Dec. 2011) using Bowtie2 (v2.4.2)[86] [parameters paired-end options (-X/−maxins 1000, --fr, --dovetail), --very-sensitive]. The aligned reads were filtered using BamTools Filter (v2.4.1)[87] (parameters --isProperPair true, --mapQuality ≥ 30, and --reference! = chrM) and the duplicate reads were removed using Picard MarkDuplicates (v2.18.2.2) (http://broadinstitute.github.io/picard/). The resulting BAM files were converted to BED and the reads that overlap to the blacklisted regions listed in https://www.encodeproject.org/files/ENCFF547MET/ were removed using Bedtools (v2.30.0)[88]. The filtered BED files were used as inputs for peak calling. Peaks were called from individual replicates using MACS2 callpeak (v2.1.1)[89] (parameters --format single-end BED, --nomodel, --extsize 200, --shift -100, --qvalue 0.05).

BAM files were then converted to BigWig with "deeptools"[90] normalizing to 1x effective genome size (2652783500 for GRCm38/mm10). BigWig files were used for data visualization in IGV with the addition of the conservation scoring by phyloP (phylogenetic *p*-values)[91] for 60 vertebrate genomes from the UCSC genome browser (mm10.60way.phyloP60way.bw file downloaded from http://hgdownload.cse.ucsc.edu/goldenpath/mm10/phyloP60way/) and ChIP-seq data from Hoxa13 (GSE81356)[60] and Gli3 (GSE133710, https://www.ncbi.nlm.nih.gov/geo/query/acc.cgi?acc=GSM3923170)[61].

The raw count matrix was created by merging peaks across samples as in refs. 92,93. DESeq2 object was built from the raw count matrix and blind dispersion estimated with the "vst" function from the "DESeq2" package in R[94]. Top 2500 most variable peaks estimated with the "rowVars" function in R were selected for the principal component analysis (PCA). PCA was performed using the built-in "prcomp" function in R. 500 top loadings contributing to PC1 and PC2 were annotated using ChIPseeker package in R[95]. Hierarchical clustering was performed on the Z-score normalized reads. Correlation distances between the peaks were computed with "as.dist" function and clustering was performed using complete linkage method with "hclust" function. Clusters

were defined using a threshold h = 0.6. For statistical analysis raw reads were analyzed using EdgeR[96]. ANOVA-like test was performed on all samples and pairwise comparisons between the samples were made by analyzing individual contrasts. Venn diagrams were created from the lists of selected differentially accessible peaks with "eulerr" package in R. Heat Maps were made from data frames containing log2 normalized counts across samples with the "pheatmap" package (https://CRAN.R-project.org/package = pheatmap).

## Footprinting analysis of ATAC-Seq data

We further used the program RGT HINT-ATAC[77] and motif matching and enrichment to identify, plot and compare transcription factors (TF) footprints in the different samples. In detail, the BAM files containing the filtered aligned reads for each biological replicate were merged and used as a matrix for the footprinting analysis for the regions corresponding to the peaks called by MACS2 (BED file) (described above). The BAM and BED files for the peaks called in the control GT and the subset of the BAM and BED files corresponding to the regions that were found accessible only in GT (pattern 1 peaks) were used for the footprint analysis and then tested for motif matching to the Hocomoco database[97] for finding the motif-predicted binding sites (mpbs) independently. Motif enrichment analysis was performed on the footprinting results for the peaks of pattern 1, using as a background the footprinting results for all the peaks called in control GT. Position frequency matrices for Lef1 (MA0768.2) and Gata3 (MA0037.4) binding motifs in *Mus musculus* were obtained from Jaspar database.

## Reporting summary

Further information on research design is available in the Nature Portfolio Reporting Summary linked to this article.

# Data availability

The sequencing data obtained in this work has been deposited in the GEO repository, accession number GSE231592 [https://www.ncbi.nlm.nih.gov/geo/query/acc.cgi?acc = GSM7291329]. In addition, we also used the following published data: conservation scoring by phyloP (phylogenetic p-values)[91] for 60 vertebrate genomes from the UCSC genome browser [http://hgdownload.cse.ucsc.edu/goldenpath/mm10/phyloP60way/] and ChIP-seq data from Hoxa13 (GSE81356)[60] and Gli3 (GSE133710 [https://www.ncbi.nlm.nih.gov/geo/query/acc.cgi?acc=GSM3923170])[61]. The reference mouse genome used in this work was GRCm38/mm10. Source data are provided with this paper.

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

## Acknowledgements
We would like to thank the members of the Mallo lab, especially Ana Casaca for continuous support at different stages of this project, the IGC mouse facility for their help with animal housing, histopathology facility for their help with histology procedures, and genomics facility for helping with the ATAC-seq procedures. This project was funded by Fundação para a Ciência e a Tecnologia (FCT) grants 2022.01629.PTDC (doi: 10.54499/2022.01629.PTDC) to MM, PTDC/BII-BTI/32375/2017 to GGM, and PhD fellowships PD/BD/128437/2017 to AL and PD/BD/128426/2017 to AD, by the PhD fellowship 36/8l-D/21 from Fundação Calouste Gulbenkian to AK, and the research infrastructures PPBI-POCI-01-0145-FEDER-022122 to the Advanced Imaging Facility, and Congento LISBOA-01-0145-FEDER-022170 to the animal facility, both co-financed by Lisboa Lisboa 2020/FEDER and FCT (Portugal).

## Author contributions
Conceptualization: A. Lozovska and M.M.; formal analysis: A. Lozovska and A.K.; funding acquisition: M.M.; investigation: A. Lozovska, A.K. and M.M.; methodology: A. Lozovska, A.D. G.G.M. A.N., A. Lopes and D.F.; Writing – Original Draft Preparation: A. Lozovska and M.M.; writing – review & editing: A. Lozovska, A.K. and M.M.

## Competing interests
The authors declare no competing interests.
