## [Peer Review File · Nature Communications]

Tgfr1 controls developmental plasticity between the hindlimb and external genitalia by remodeling their regulatory landscapeREVIEWER COMMENTS

Reviewer #1 (Remarks to the Author):

In the manuscript by Lozovska et al, the authors present the phenotype of conditional loss of Tgfb1 function driven by Cdx2-CreERT with tamoxifen injection at embryonic day 6.75. Whereas Tgfb1 null embryos fail to initiate hindlimb or cloacal structures, conditional loss of Tgfb1 in this strategy produced embryos with urogenital defects and duplicated hindlimbs. Gene expression analysis is consistent with a transformation of pericloacal to hindlimb fate. The authors then conduct an ATAC-Seq analysis of chromatin accessibility in control hindlimb and genital tubercle and of the extra hindlimbs and pericloacal tissues of Tgfb1-cKO embryos and identify patterns of differential accessibility suggesting the pericloacal tissue accessibility pattern has been partially transformed to a more hindlimb-like pattern.

The mutant phenotype is very interesting in the context of prior work that shows the shared developmental history of hindlimbs and the genital tubercle. Evidence for the role of Tgfb1 to segregate and restrict tissue identities at later stages substantially adds to the field. However, there are some points that need to be addressed prior to publication.

1) Tgfb1 needs to be clearly defined as the 'Transforming growth factor receptor b1' at its first mention in the text to aid a novice reader in understanding the findings and their interpretation. This is important, because the text also needs to be modified to make it very clear that the chromatin responsiveness is an indirect outcome of Tgfb1 receptor activity. For example 'Tgfb1 controls the response to those factors acting in a pioneer like mode to modulate...' in the abstract could be read to suggest that Tgfb1 itself acts as a pioneer. Another example is in the Discussion: "Tgfb1 controls pericloacal mesoderm plasticity by modulating its regulatory landscape" suggests this could be direct. I think simply stating 'Tgfb1 signaling' or 'factors downstream of (or influenced by) Tgfb1 signaling' throughout the manuscript will make this more clear.

2) The chromatin response to loss of Tgfb1 is complex. Some sequences require Tgfb1 signaling for their accessibility in the GT (Pattern 1). Other sequences seem to require Tgfb1 signaling to enable their accessibility in the hindlimb (Pattern 3). However, other sequences seem to require Tgfb1 signaling to restrict their accessibility in pericloacal tissue, perhaps such that it doesn't become limb (Pattern 2). Because of this last group, and because the direct chromatin modifier(s) have not yet been identified, I don't think it's necessary or appropriate to invoke 'pioneer activity'. The authors seem to recognize this, since they indicate a couple of times in the discussion that the Pattern 2 elements complicate this overly simple interpretation. I think it's sufficient to say that Tgfb1 signaling modulates the regulatory landscape and then discuss how this may or may not work (e.g., Smads etc). I also wouldn't use the term 'locked' and would instead say that the chromatin is inaccessible. Locked suggests an irreversible state that can't be known in this snapshot experiment.

3) Supplementary Figure 1 is confusing. Are Panels B-D showing the phenotype, transcript, and protein production of Tgfb1^{3ex3} animals in absence of Cre activity? This should all be made more clear by

noting whether or not the 3ex3 allele is recombined in these panels. And then Panel E focuses on an allelic series using only the floxed single exon 3 allele with and without Cre activation? Given the authors state that they used the 3ex3 allele for much of the paper due to its higher sensitivity to tamoxifen, please add normalized *Tgfb1* expression in these animals (recombined and not recombined) to panel E. As is, the figure doesn't clearly show how this allele produces a loss of *Tgfb1* function but rather seems to focus on its phenotype in absence of Cre (if I'm reading this correctly).

4) I am fascinated by the fact that the extra pair of hindlimbs in Figure 1C (magenta) form entirely independent of the endogenous hindlimbs and with such 'normal' proximal distal and anterior posterior patterning given the gene expression patterns of the enlarged hindlimb/pericloacal structures seen in Figure 2. For example, looking at *Fgf8* expression in Figure 2A', I might have expected a single limb to form, perhaps with polydactyly because of the excessive anterior/posterior expanse. These are, however, young embryos. The older embryo stained with *Shh* in Supplementary Figure 2D' seems to hint at segregation of two distinct outgrowths, and I see two posterior *Shh* signaling centers. To better understand how the expanded hindlimb field might segregate into two distinct limbs, please show later (E11.5-E12.5) expression of *Fgf8* and *Hand2* as well as an anterior limb marker, such as *Alx4* or *Pax9*.

Reviewer #2 (Remarks to the Author):

In this paper the authors explore the capacity of the early primordium of the mouse external genitalia to make limbs. Through the genetic removal of *Tgfb1*, the authors uncover the remarkable plasticity of this structure. They generate two floxed *Tgfb1* alleles and use the *Cdx2CreERT* line to conditionally remove this receptor. Mice with the conditional removal exhibit omphalocele and hindlimb duplication. In these mutants, WMISH reveals that the pericloacal region shows gene expressions normally observed in the hindlimb (HL) supporting a conversion of the pericloacal region into HL. Furthermore, the authors employ ATAC-seq to analyze changes in chromatin accessibility in the pericloacal region, which they classify into three distinct patterns. The authors conclude that *Tgfb1* controls the identity of the pericloacal tissue by modulating its regulatory landscape.

This is an interesting study; however, my overall opinion is that it is presented in a too compacted and occasionally cryptic manner. One major concern is that the conditional removal of the receptor occurs very early (Tam administration at E6.75 and E7.25) but the analyses are performed several days later (E11.5 for the ATAC-seq) raising the possibility that the effects are not direct. This is further suggested by the absence of *Smad* binding sites in the differentially accessible regions. Therefore, in my opinion, the possibility that the changes in chromatin accessibility observed are indirectly caused by *Tgfb1* need to be considered.

Other issues that require clarification are listed here:

1. *Tgfb1* is conditionally removed using the *Cdx2-creERT* line. It would be necessary to show the activity of *Cdx2-creERT* line (e.g. crossing with *ROSA26R* or other reporter line) at the time of Tam administration in this study. Alternatively, referencing a publication that shows this activity would be appropriate. It is

also important to describe the expression pattern of *Tgfb1* and overlapping with the activity of the deleter line.

2. In the Introduction, please provide basic background on *Tgfb1* function, structure of the receptor, pathways, etc..).

3. The description of the mutant phenotype requires improvement:

- describe and discuss the morphologic variability of the extra HLs. Include skeletal staining of at least one representative mutant HL. This is crucial for understanding the impact of the fragmented and distal expression of *Shh*. The current segmentation images (Fig. 1C) do not fully reveal the pattern of skeletal elements and digits.

- Clarify the timing of the phenotype onset, including at the internal level. Explain the nature of the protrusions found in the mutant cloaca and whether the authors can trace them in their OPT analyses.

4. The ATAC-seq studies also require improvement:

- Please include a scheme depicting the dissected regions.

- Fig. 3C shows that the chromatin accessibility of the mutant extralimb is closer to the mutant and control GTs in the first dimension, as indicated in the text (lines 157-158) but it also shows that in the second dimension the mutant HL and the mutant GT are closer to the limb than to the control GT. Could the authors explore and compare the topmost variable accessible regions in both the first and second dimensions? This comparison may provide valuable insights into the control of limbness versus pericloacal tissue.

- Regions with different accessibility between samples are being considered. Please provide the fold change and adjusted p-value to assess the significance. When do the authors consider that one region is locked?

- Pattern 2 and, particularly, Pattern 3 should receive more attention/explanation in the text. The heatmap in Fig. 4A' includes 2031 regions, while the heatmap in Fig. 4B' includes 13471 regions. Could the authors check if this is accurate, as the thickness of the lines looks similar?

- Please provide more information about the *Hox13* and *Gli3* ChIP-seq tracks. Are the peaks shown identified through peak calling? are the *Gli3* peaks referred to as responsive to *Shh*, as described in Lex et al. 2020 (PMID: 31989924)?

5. The footprinting analysis should be included in the results section. Given that *Smad* binding is not found, maybe the authors would like to explore the motif enrichment in the elements putatively regulated by *Tgfb1*.

6. Is *Tgfb1* transducing *Bmp* signaling? *Bmps* has been reported to be necessary for the formation of the aural/ventral mesoderm (PMID: 15843411; PMID: 23028704)

Minor:

- Please, separate (and mark) the Introduction and Results sections.

- Considering the early administration of tamoxifen (at E6.75 and E7.25), why do the authors set their objective in understanding the role of the receptor in later stages of HL development?

- There is some confusion between the pericloacal region and the GT- For example, the ATAC-seq is performed in extra limbs and pericloacal region (line 155) but in the figure captions the samples are considered GT. However, in other places in the text the pericloacal and GT are considered as two different tissues. Please clarify thought-out the text.

- The rationale behind the *Tgfb13ex3-flox* allele requiring half the tamoxifen dosage compared to the *Tgfb1ex3-flox* allele for the same phenotype needs further clarification.

- Why not performing the immunoblot analysis with the embryo tissue? is the higher expression level of *Tgfb13ex3* in HEK cells consistent?
- Please, indicate in the text the stages, number of samples analyzed in each experiments (WMISH), etc...

Reviewer #3 (Remarks to the Author):

The MS NCOMMS-23-25532-T described on the unique mouse duplicated HL (hindlimb) phenotype by modulating TGF β signals. The duplication phenotype will appeal not only to genital researchers but readers for limb, appendage programs also for evolutionary viewpoints.

Contrary to the merits of phenotypes, there are problems about the

1 mechanistic analyses for the phenotypes and 2 lack of future perspectives in the MS.

About the mechanisms, the main issue is whether they can show convincing data for the fate change; how TGF signals change fates from GT to limbs.

The MS basically shows marker analyses and regulatory element informatics.

Marker data such *Shh* signals, *Isl* (although not so clean), *Bmps* are basically in agreement with the “ changes “ but not explaining the mechanisms.

They mentioned several element-perspectives such as ZRS (for HH signals) and *Isl*-related elements.

But they are again just speculations.

If they suggest the involvement of Smads for TGF mediated fate change, its interesting but it is not the case.

Why TGF-(and not by smad) as “ pioneer regulator “ remains totally unclear which they also state in Discussion.

They described *Tgfb1/BMP* signaling pathways are not among the main regulators of the regulatory elements and can not suggest other candidate signals.

Other points;

They mentioned little on *Bmp* for early peri cloacal field. There are already works reporting mermaid (Sirenomelia) phenotype in which such field is reduced leading to HL fusion.

It was originally shown in Bmp/Tsg mutant (PMID: 15843411 DOI: 10.1242/dev.01822) and later shown by conditional Bmp KO by Isl cre (journal.pone.0043453. Epub 2012 Sep 17. PMID: 23028455). Hence, if Tgf mediated modulation affects (possibly by Smad ?) to Bmp signal, it is interesting. But it is not the case.... They should be cited. In fact, the current urogenital phenotypes include lack of kidney and small bladder and such regions are again affected in such Isl Cre Bmp mutants.

The initial Fgf -GTpaper (2000) is not cited.

Why Cdx cre line? What are the roles of such Cdx posi cells and TGF signals in such cells? They did not describe on it. The authors performed significant works on axial elongation before. The Cdx population of cells are such cells or utilized just for broad cells in early posterior embryos? As above, if they use Isl cre for TGF modulation it can be interesting.

NPG (Nature groups) published a short comment type work for ext genitalia (the short one for Hox ,HFGS which they cited). Judged by the unique-strong HL phenotypes, I wonder the MS can go for commentary or matters rising sections etc if they want NC publication. The MS can go for Comm Bio, which can list much data for the science community.

Reply to reviewers

Reviewer 1

We thank this reviewer for his/her comments, which clearly helped to improve our manuscript. We would also like to state that the short introduction and rather compacted description of the data in the original manuscript, which often led to the absence or incomplete information, resulted from the history of our submission, initially sent to Nature (and thus adjusting to its more compacted style), which was directly transferred to Nature Communications keeping the original format. In the revised manuscript we have, in addition to addressing the specific comments, substantially expanded the manuscript, including more background, a clearer (we hope) description of our data and a more balanced discussion of their implications.

1) Tgfbr1 needs to be clearly defined as the ‘Transforming growth factor receptor b1’ at its first mention in the text to aid a novice reader in understanding the findings and their interpretation. This is important, because the text also needs to be modified to make it very clear that the chromatin responsiveness is an indirect outcome of Tgfbr1 receptor activity. For example ‘Tgfbr1 controls the response to those factors acting in a pioneer like mode to modulate...’ in the abstract could be read to suggest that Tgfbr1 itself acts as a pioneer. Another example is in the Discussion: “Tgfbr1 controls pericloacal mesoderm plasticity by modulating its regulatory landscape” suggests this could be direct. I think simply stating ‘Tgfbr1 signaling’ or ‘factors downstream of (or influenced by) Tgfbr1 signaling’ throughout the manuscript will make this more clear.

We introduced the extended name Transforming growth factor receptor b1 in the introduction (lines 85-86). We agree with the reviewer that, while it is clear that the global regulatory landscape of the pericloacal mesoderm, as defined by accessibility patterns, is different in the presence or absence of Tgfbr1 signaling, we cannot explain how this is achieved. We agree that the pioneer activity (which would imply some direct interaction between pathway components and the regions under regulation) is unlikely to provide a satisfactory explanation. We thus modified the text removing all references to a possible pioneering activity and to avoid too much speculation about specific mechanisms.

2) The chromatin response to loss of Tgfbr1 is complex. Some sequences require Tgfbr1 signaling for their accessibility in the GT (Pattern 1). Other sequences seem to require Tgfbr1 signaling to enable their accessibility in the hindlimb (Pattern 3). However, other sequences seem to require Tgfbr1 signaling to restrict their accessibility in pericloacal tissue, perhaps such that it doesn’t become limb (Pattern 2). Because of this last group, and because the direct chromatin modifier(s) have not yet been identified, I don’t think it’s necessary or appropriate to invoke ‘pioneer activity’. The authors seem to recognize this, since they indicate a couple of times in the discussion that the Pattern 2 elements complicate this overly simple interpretation. I think it’s sufficient to say that Tgfbr1 signaling modulates the regulatory landscape and then discuss how this may or may not work (e.g., Smads etc). I also wouldn’t use the term ‘locked’ and would instead say that the chromatin is inaccessible. Locked suggests an irreversible state that can’t be known in this snapshot experiment.

As discussed in the reply to the previous point, we agree with the reviewer that the use of “pioneering activity” might not be appropriate. We tried to avoid too much speculation about the generation of the different accessibility patterns in the presence and absence of *Tgfb1* and centered our discussion on the consequences of those differences to the control of the differentiation fate of the pericloacal mesoderm. In this regard, and following reviewer 2 suggestion, we now describe the footprinting data in the results section (lines 400-452) (it was just a note in the discussion section of the original manuscript) including a proof of principle experiment to provide some support to our hypothesis (Fig. 6 and supplementary Fig. 4). Also, following the reviewer suggestion, we changed the word “locked” for inaccessible.

*3) Supplementary Figure 1 is confusing. Are Panels B-D showing the phenotype, transcript, and protein production of $Tgfb1^{3ex3}$ animals in absence of Cre activity? This should all be made more clear by noting whether or not the 3ex3 allele is recombined in these panels. And then Panel E focuses on an allelic series using only the floxed single exon 3 allele with and without Cre activation? Given the authors state that they used the 3ex3 allele for much of the paper due to its higher sensitivity to tamoxifen, please add normalized *Tgfb1* expression in these animals (recombined and not recombined) to panel E. As is, the figure doesn't clearly show how this allele produces a loss of *Tgfb1* function but rather seems to focus on its phenotype in absence of Cre (if I'm reading this correctly)*

Supplementary Figure 1, panels B and B' show stained skeletons of a control (B) and a trans-heterozygote for the null and the *Tgfb1^{3ex3}* alleles (not recombined; the recombined allele would give the extra hindlimb). This is to show that the *Tgfb1^{3ex3}* allele generates a product that retains some activity (the embryos develop much further than the *Tgfb1* nulls) but less active than the wild type allele (*Tgfb1* heterozygotes are normal). Panels C and D show that the *Tgfb1^{3ex3}* allele generates a spliced mRNA of increased size (C) (RT-PCR using RNA obtained from *Tgfb1^{3ex3}* homozygous embryos), and that this RNA is translated into the protein larger than wild type (expressed in cells) (D). We included a more straightforward description of this allele and clearer explanation of the figures in the first section of result (lines 134-145) and in the figure legend. Also, as suggested by the reviewer, we added to the figure an additional panel (panel G) showing the normalized expression of *Cdx2Cre^{ERT};Tgfb1^{3ex3}-* embryos.

*4) I am fascinated by the fact that the extra pair of hindlimbs in Figure 1C (magenta) form entirely independent of the endogenous hindlimbs and with such 'normal' proximal distal and anterior posterior patterning given the gene expression patterns of the enlarged hindlimb/pericloacal structures seen in Figure 2. For example, looking at *Fgf8* expression in Figure 2A', I might have expected a single limb to form, perhaps with polydactyly because of the excessive anterior/posterior expanse. These are, however, young embryos. The older embryo stained with *Shh* in Supplementary Figure 2D' seems to hint at segregation of two distinct outgrowths, and I see two posterior *Shh* signaling centers. To better understand how the expanded hindlimb field might segregate into two distinct limbs, please show later (E11.5-E12.5) expression of *Fgf8* and *Hand2* as well as an anterior limb marker, such as *Alx4* or *Pax9**

We performed additional *in situ* hybridization experiments on E11.5 embryos, including *Fgf8* (results, lines 222-226 and Fig. 3A'), *Hand2* (results, lines 246-250 and Fig. 3B'), and *Pax9* (results, lines 250-252 and Fig. 3C'), in which, in addition to the expression, the separation of the two hindlimb prominences can be appreciated.

The question of the segregation of the apparently single hindlimb field observed at E10.5 into two independent limb structures is very interesting. Experiments aimed to address one of the comments from reviewer 2 provided the basis to propose the origin of such segregation. In particular, close evaluation of the two components of the principal component (PC) analysis of the ATAC-seq samples revealed that, while one of them (PC2) showed similarities between the two hindlimb samples, the other (PC1) indicated that the extra hindlimb region clustered together with the other samples obtained from the region around the cloaca (both wild type and mutant) and not with the normal hindlimb. This might indicate that the tissue derived from the pericloacal mesoderm and the prospective limb regions contain different cell adhesion properties that would prevent them from mixing (this would be indicated by PC1), properties that might still be present in the tissue in the absence of *Tgfb1*. In the absence of mixing, the activation of a hindlimb program in this tissue (likely revealed by PC2 patterns) would produce two independent limb structures. We introduced these data in the revised manuscript (lines 290-300 and Fig. 4F,G), as well as its possible implications for the evolution of the pericloacal/prospective hindlimb area in squamates (lines 458-471).

Reviewer 2

We would also like to thank this reviewer for his/her comments, as they clearly helped to improve our manuscript. Similarly to what we wrote in a preliminary comment to reviewer 1, here we would also like to note that several of the deficiencies that triggered the comments of this reviewer derived from the fact that this manuscript was directly transferred from our initial submission to Nature and, therefore, its structure was fitting to Nature's format. We have now rewritten the introduction and the discussion and extended the description of our data to fit Nature Communications format, which also helped to address several of this reviewer's comments.

1) Tgfb1 is conditionally removed using the Cdx2-creERT line. It would be necessary to show the activity of Cdx2-creERT line (e.g. crossing with ROSA26R or other reporter line) at the time of Tam administration in this study. Alternatively, referencing a publication that shows this activity would be appropriate. It is also important to describe the expression pattern of Tgfb1 and overlapping with the activity of the deleter line.

The *Cdx2-cre^{ERT}* line had already been described in Jurberg et al, 2013 (doi: 10.1016/j.devcel.2013.05.009, reference also provided in the manuscript) and used in Arias et al, 2019 (doi: 10.1016/j.devcel.2018.12.004) and de Lemos et al, 2022 (doi: 10.1242/dev.198812). We now added a picture of the ROSA26R-Bgal reporter activated by *Cdx2-Cre^{ERT}* upon tamoxifen administration at E7.5 (supplementary figure 1E) to show the embryonic region where *Tgfb1* is expected to be inactivated.

We repeatedly attempted to visualize *Tgfb1* localization, both mRNA (including HCR) and protein, *in vivo* at mid gestation stages relevant for our study. We were

unable to confirm the expression described in the single report showing *Tgfbr1* expression in mid-gestation mouse embryos (doi: 10.1038/sj.embor.7400752). Instead, our experiments seemed to show ubiquitous low-level expression, requiring long developing times, and difficult to differentiate from background (an example shown in the picture below). This was the main reason we decided to estimate the efficiency of the inactivation relying on quantitative RT-PCR as shown in supplementary Fig. 1.

Figure: *in situ* hybridization on a wild type E9.5 embryo using a *Tgfbr1* anti-sense probe.

2) *In the Introduction, please provide basic background on Tgfbr1 function, structure of the receptor, pathways, etc..)*

As explained in the initial note, we have now extended the introduction, which includes the requested background. *Tgfbr1* function and pathways are outlined in lines 85-114 and 401-406, and relevant features of *Tgfbr1* structure are now included in the results (lines 140-142).

3) 3. *The description of the mutant phenotype requires improvement:*
- *describe and discuss the morphologic variability of the extra HLs. Include skeletal staining of at least one representative mutant HL. This is crucial for understanding the impact of the fragmented and distal expression of Shh. The current segmentation images (Fig. 1C) do not fully reveal the pattern of skeletal elements and digits*

In preliminary experiments we had performed skeletal staining of one *Tgfbr1-cKO* mutant (an image is shown below). Maybe because it was difficult to properly remove the skin from the limbs without damaging them, which interfered with the penetration of the alcian blue stain, together with the general fragility of those mutant fetuses, it was not possible to obtain relevant images from the hindlimb area. For that reason, we decided to perform the OPT analyses. To respond to the reviewer's comment, we now included images that had been obtained with high resolution light sheet microscopy, that allowed detailed segmentation of the skeletal structures of the mutant limb autopods. These data are included in supplementary Fig. 2.

Figure: Staining of a *Tgfr1-cKO* fetus using the standard alcian blue/alizarin red method. The arrows indicate the two sets of hindlimbs.

- Clarify the timing of the phenotype onset, including at the internal level. Explain the nature of the protrusions found in the mutant cloaca and whether the authors can trace them in their OPT analyses.

In preliminary experiments, we could not see any clear phenotype at E9.5, most likely because limbs are just being induced at this stage, and the GT primordium is induced one day later, at E10.5. We therefore did not perform in situ analyses at E9.5. We did some analysis at early E10.5 (for instance, *Fgf8*, *Lin28a*, *Fgf10*), which we believe is the most adequate for early limb development and already provided relevant information (Figure 2).

The protrusions in the endodermal cloaca were already evident around E10.5 (e.g. supplementary Fig. 3E'). OPT data did not have enough resolution to trace cloacal protrusions. However, we observed what could be some remains of those protrusions by conventional histological sections (e.g. supplementary Fig. 3C').

4) The ATAC-seq studies also require improvement:

- Please include a scheme depicting the dissected regions

- We included a scheme in Fig. 4C.

- Fig. 3C shows that the chromatin accessibility of the mutant extralimb is closer to the mutant and control GTs in the first dimension, as indicated in the text (lines 157-158) but it also shows that in the second dimension the mutant HL and the mutant GT are closer to the limb than to the control GT. Could the authors explore and compare the topmost variable accessible regions in both the first and second dimensions? This comparison may provide valuable insights into the control of limbness versus pericloacal tissue.

We thank the reviewer for this very interesting suggestion. Indeed, the suggested analysis led us to formulate a hypothesis to explain the generation of two independent limb structures despite the presence of a single extended hindlimb bud in early *Tgfr1-cKO* mutant embryos.

We explored top 500 most variable peaks in PC1 and PC2. We annotated those peaks, compared distribution in the genome between PC1 and PC2 top peaks and performed their hierarchical clustering (HC). HC showed clusters of peaks potentially contributing to the limb phenotype from the pericloacal region of the mutants (cluster 1 and 3 in PC2). Interestingly, PC1 showed that the extra hindlimb region clustered together with the other samples obtained from the region around the cloaca (both wild type and mutant) and not with the normal hindlimb. This might indicate that the tissue derived from the pericloacal mesoderm and the prospective limb regions contain different cell adhesion properties that would prevent them from mixing (this would be indicated by PC1), which might be still present in the tissue in the absence of *Tgfb1*. In the absence of mixing, the activation of a hindlimb program in this tissue (likely revealed by PC2 patterns) would produce two independent limb structures. We introduced these data in the revised manuscript (lines 283-300 and Fig. 4F,G), as well as its possible implications for the evolution of the pericloacal/prospective hindlimb area in squamates (lines 458-471).

- Regions with different accessibility between samples are being considered. Please provide the fold change and adjusted p-value to assess the significance. When do the authors consider that one region is locked?

We selected differential peaks with threshold of $FDR < 0.001$ and $\log FC > 1.5$ (indicated in the legends for the corresponding figures and in the text). Also, as was suggested by reviewer 1, we avoided using termed “locked” in the text and referred to reduced accessibility instead.

- Pattern 2 and, particularly, Pattern 3 should receive more attention/explanation in the text. The heatmap in Fig. 4A' includes 2031 regions, while the heatmap in Fig. 4B' includes 13471 regions. Could the authors check if this is accurate, as the thickness of the lines looks similar?

The thickness of the lines in the heatmaps seems similar because to build the figure, the heatmaps were generated to match the same final size. These heatmaps include too many peaks to distinguish individual lines, some of them are probably merged together to fit into a graph of given dimensions.

- Please provide more information about the Hox13 and Gli3 ChIP-seq tracks. Are the peaks shown identified through peak calling? are the Gli3 peaks referred to as responsive to Shh, as described in Lex et al. 2020 (PMID: 31989924)?

As specified in the manuscript, these tracks were taken from published datasets, and loaded directly into IGV using the default parameters that revealed the presence or absence of distinct peaks. The same parameters were used in Figs 4 I,J,K, where it shows that in the ChIP-seq analyses of the forelimb bud neither *Hoxa13* or *Gli3* relevant binding was detected in those GT-specific peaks, and in Fig. 5C,D,F, corresponding to enhancers accessible in the normal hindlimb, where *Hoxa13* and *Gli3* binding was found enriched in the forelimb bud. *Gli3* peaks are indeed referred to as potentially *Shh*-responsive, similarly to what was described in Malkmus et al 2021 (referenced in our manuscript), which is where the enhancers involved in *Grem1* regulation by *Shh* were described.

5) *The footprinting analysis should be included in the results section. Given that Smad binding is not found, maybe the authors would like to explore the motif enrichment in the elements putatively regulated by Tgfbr1.*

We now included the footprinting analysis in the last section of results (lines 400-452), exploring motif enrichment in the accessible elements specific for the control GT (thus likely accessible only in the presence of *Tgfbr1*). We also performed a proof of principle experiment by analyzing a Lef1 footprint in the potential *Isl1* element identified as one of the pattern 1 elements (shown in Fig. 6 and supplementary Fig 4). The results from these experiments are consistent with its involvement on the regulation of this enhancer, thus fitting with our general interpretation of the *Tgfbr1-cKO* phenotype.

6) *Is Tgfbr1 transducing Bmp signaling? Bmps has been reported to be necessary for the formation of the caudal/ventral mesoderm (PMID: 15843411; PMID: 23028704)*

Background on BMPs involvement in formation of the caudal body and its potential involvement in the activation of *Tgfbr1* was now included in introduction, lines 104-114.

Minor:

- Please, separate (and mark) the Introduction and Results sections.

As indicated in the preliminary note, we reformatted the manuscript which included separated Introduction and results sections.

- Considering the early administration of tamoxifen (at E6.75 and E7.25), why do the authors set their objective in understanding the role of the receptor in later stages of HL development?

We agree that this does not sound logic. Actually, the initial objective of these experiments was to explore the role of *Tgfbr1* in spinal cord development. For this we had to bypass the lethality of the *Tgfbr1* null mutant, allowing the embryo to grow beyond the trunk to tail transition. Considering the dynamics of cre activity upon tamoxifen administration from previous work in the lab, we performed some initial experiments which showed the remarkable phenotype described in this paper, which left us no option but to change the objective and follow the analysis of the limb phenotype (although not shown in this paper, the spinal cord is also larger than in control embryos). The final timing and dosage for tamoxifen administration was then further refined in subsequent experiments to obtain the parameters used in this work.

- There is some confusion between the pericloacal region and the GT- For example, the ATAC-seq is performed in extra limbs and pericloacal region (line 155) but in the figure captions the samples are considered GT. However, in other places in the text the pericloacal and GT are considered as two different tissues. Please clarify thought-out the text.

We clarified ATAC-seq samples and their nomenclature in lines 274-276 and tried to modify the text to avoid possible confusion.

- The rationale behind the *Tgfr13ex3-flox* allele requiring half the tamoxifen dosage compared to the *Tgfr1ex3-flox* allele for the same phenotype needs further clarification.

Based on the genetic analyses, we believe that *Tgfr1^{3ex3}* encodes a hypomorphic receptor. For instance, trans-heterozygotes between *Tgfr1^{3ex3}* and a *Tgfr1* null allele develop way past the null mutants (*Tgfr1* null mutants become arrested at E10.5) but, contrary to *Tgfr1* heterozygotes, were unable to give life progeny, generating a phenotype resembling in many respects the *Gdf11* null mutants. If cre-mediated deletion upon tamoxifen administration is not 100% efficient, it is likely that deletions in *Tgfr1^{3ex3}* reach the functional threshold that does not support *Tgfr1* signaling activity easier than when using *Tgfr1^{flox}*. We added more detailed description of this allele to the first section of the results (lines 131-145) together with the data in supplementary Fig. 1.

- Why not performing the immunoblot analysis with the embryo tissue? is the higher expression level of *Tgfr13ex3* in HEK cells consistent?

We had originally tried to perform the Western blot experiment using protein extracts from homozygous *Tgfr1^{3ex3}* embryos and wild type control, additionally using *Tgfr1* null embryos as negative controls. In those experiments we had found that the antibody cross reacts with other proteins *in vivo* (see for instance an example of such experiment, which actually questions its use in immunofluorescence experiments). We obtained discrete bands from the same membrane with the control hybridization against actin, thus showing that it was not a problem with the transfer to the membrane. The images did not improve when changing conditions in the Western assay. With such background, even in the lane of containing the extract from null mutants, it was not possible to identify the bands corresponding to the proteins we wanted to characterize (even the band that seems to disappear in the null lane is bigger than the expected size of *Tgfr1*). As the antibody had been shown to give clean images when tested in transfected cell lines (maybe because its production in excess), we decided to isolate mRNA from the wild type and *Tgfr1^{3ex3}* homozygous embryos, clone the cDNAs into expression vectors, express them in cell lines and identify the resulting products by Western blot. These experiments showed that the mRNA produced by the *Tgfr1^{3ex3}* allele can indeed generate a protein with higher molecular weight than the wild type receptor. The higher expression level shown in the blot of supplementary Fig. 1 might have to do with transfection efficiency but should not represent a difference in expression in the embryo.

- Please, indicate in the text the stages, number of samples analyzed in each experiments (WMISH), etc...

The stages are now indicated in the text and figure legends. The number of embryos analyzed for each probe and stage are also indicated in the corresponding figure legends.

Reviewer #3 (Remarks to the Author):

As already also explained to the other reviewers, some of the deficiencies highlighted by this reviewer were the consequence of the compressed version of the original manuscript, more fitting a Nature format (the initial submission), that was directly transferred from Nature to Nature Communications, without a change in format.

The MS NCOMMS-23-25532-T described on the unique mouse duplicated HL (hindlimb) phenotype by modulating TGFb signals. The duplication phenotype will appeal not only to genital researchers but readers for limb, appendage programs also for evolutionary viewpoints.

Contrary to the merits of phenotypes, there are problems about the

1 mechanistic analyses for the phenotypes and 2 lack of future perspectives in the MS.

1) We disagree that this manuscript is just descriptive lacking mechanistic analyses. Indeed, we have uncovered a new mechanism by which Tgfbr1 controls the morphogenesis of the pericloacal tissue to enter GT fates. We show that in this case Tgfbr1 does not follow any of the canonical mechanisms typically associated with members of the Tgfb/BMP signaling family, thus increasing the relevance of our findings. Indeed, instead of regulating the activity of specific genes through the activation of Smad responsive elements as typically associated with the Tgfb/BMP pathways, our ATAC-seq data show that it determines the regulatory regions accessible to the control by relevant morphogenetic factors (Shh effectors, Hox13 proteins, etc) in the specific tissue. We now included additional data, also involving a proof of concept experiment (lines 423-451, Fig. 6 and supplementary Fig. 4) that supports our interpretation that Tgfbr1 activity renders regulatory regions accessible to other regulatory factors controlling their activity. This is also supported by our data with the *Gremlin1* enhancer. This is now further explained in the discussion (lines 514-525).

2) The future perspectives are now more explicit in the revised manuscript. In summary they involve:

a- The search for the mechanism(s) mediating this novel Tgfbr1 activity, meaning, how Tgfbr1 promotes global control of the regulatory landscape in the pericloacal mesoderm. This can be read in the discussion (lines 544-552): "A challenging question left open by our work is deciphering the mechanism(s) by which Tgfbr1 activity controls such large-scale remodeling of the regulatory landscape in the target

tissues. The lack of enrichment in footprints for Tgfb1 effectors in the target regions makes a direct activity of the canonical Tgfb1 pathway in this process unlikely. Instead, it is more probable that this regulation is indirect. Whatever the mechanism or mechanisms, it requires a high degree of coordination, which would still be more challenging if several mechanisms are involved in the generation of the different patterns (e.g. if the control of accessibility and inaccessibility relies on independent mechanisms)".

b- Understanding the evolutionary implications of our findings. Regarding this, we wrote (lines 488-491 in the discussion): "It will be therefore interesting to determine whether a mechanism related to the developmental plasticity uncovered by our work, including changes in cell affinity properties of prospective hindlimb and genital primordia, could help explaining the absence of hindlimbs in snakes but their presence in most lizards".

We further added (lines 526-532): "The identification of chromatin elements specifically associated with hindlimb and GT fates might guide the search for the molecular signature that specifies hindlimb or GT development. The large number of elements in each group suggest that these signatures might be quite complex, likely involving the combined activity of several factors. Identification of those factors will be challenging, considering that in mammals enhancers are typically located far from the genes they regulate, which are often not the closest transcription unit".

c- Understanding whether a similar mechanism operates in other processes under Tgfb1 control. Accordingly, we wrote (lines 552-556): "Identification of those mechanisms and determining whether they also operate in other physiological and pathological processes under the control of members of the Tgfb/BMP signaling family might have far reaching implications for our understanding of morphogenetic processes and disease".

About the mechanisms, the main issue is whether they can show convincing data for the fate change; how TGF signals change fates from GT to limbs.

We think that the development of an extra set of hindlimbs is a pretty strong demonstration of the fate change. It is also supported by the changes in gene expression (e.g. loss of GT markers and gain of limb markers). Indeed, in a later comment, this reviewer already acknowledges that the marker data is in agreement with the fate change. Also, our data does not show that TGF signals change fates from GT to limbs but just the opposite (the extra hindlimbs develop in the *Tgfb1* mutants) Additionally to the marker analysis we show that global chromatin state on the mutant extra limb clusters closer with both mutant and control GT, implying common origin of these structures.

The MS basically shows marker analyses and regulatory element informatics.

We not only show marker analyses and regulatory element informatics, but we analyze and interpret them in specific contexts that allow building significant conclusions.

Marker data such Shh signals, Isl (although not so clean), Bmps are basically in agreement with the " changes " but not explaining the mechanisms.

As above, the markers were used to support the fate change initially observed in the analysis of the remarkable phenotype observed in the E16.5 mutant embryos. The general mechanism was discovered through the ATAC-seq analyses, which revealed the existence of global changes in the accessibility profiles of the chromatin. We now further analyzed the ATAC-seq profiles and expanded the footprinting analysis, including a proof of principle, to further sustain that the modulation of the fate was not derived from canonical *Tgfbr1* signaling activity but involved a novel mechanism (lines 400-452 and Fig. 6 and supplementary Fig 4)

Also, we changed the *Tgfbr1-cKO* embryo stained for *Isl1*, for an embryo that better illustrate the lack of expression of this gene in the pericloacal region.

They mentioned several element-perspectives such as ZRS (for HH signals) and Isl-related elements.

If they suggest the involvement of Smads for TGF mediated fate change, its interesting but it is not the case.

We are not sure why involvement of Smads in changes would be more interesting than non involvement. Regardless of this, what we suggest is actually that canonical Smad mediated mechanisms are most likely NOT involved in the fate change.

Why TGF-(and not by smad) as “ pioneer regulator “ remains totally unclear which they also state in Discussion.

As suggested by the other reviewers, a pioneer activity would imply a more direct role on each of the regions whose accessibility is modified, which now cannot be either demonstrated or ruled out. We therefore removed any mention of a possible pioneer activity.

They described Tgfbr1/BMP signaling pathways are not among the main regulators of the regulatory elements and can not suggest other candidate signals.

We actually describe that *Tgfbr1* signaling is the main regulator of the accessibility changes in the genome (they are produced upon the inactivation of this gene). What we state is that this is not done following canonical signaling schemes for this pathway. The way this is controlled is one of the specified key questions for the future.

Other points;

They mentioned little on Bmp for early peri cloacal field. There are already works reporting mermaid (Sirenomelia) phenotype in which such field is reduced leading to HL fusion.

It was originally shown in Bmp/Tsg mutant (PMID:15843411

DOI:10.1242/dev.01822) and later shown by conditional Bmp KO by Isl cre

(journal.pone.0043453. Epub 2012 Sep 17.PMID: 23028455). Hence, if Tgf mediated

modulation affects (possibly by Smad ?) to Bmp signal, it is interesting. But it is not the case.... They should be cited. In fact, the current urogenital phenotypes include lack of kidney and small bladder and such regions are again affected in such Isl Cre Bmp mutants

The initial Fgf -GTpaper (2000) is not cited.

In the revised version we have expanded considerably the introduction, that now includes references to the roles of different signaling pathways in the morphogenesis of the GT, including (but not restricted to) the references to Fgf signaling, and to the Bmp-related mutants specified by the reviewer (lines 67-74 and 104-114).

Why Cdx cre line? What are the roles of such Cdx posi cells and TGF signals in such cells? They did not describe on it. The authors performed significant works on axial elongation before. The Cdx population of cells are such cells or utilized just for broad cells in early posterior embryos? As above, if they use Isl cre for TGF modulation it can be interesting.

The *Cdx2* enhancer we used in this work is active in all tissues that will generate the hindlimb and external genitalia as we had already shown in Jurberg et al 2013 (doi: 10.1016/j.devcel.2013.05.009, reference also provided in the manuscript). The use of a tamoxifen induced version of cre allowed temporal control of the recombination, which was important to bypass other previous functions of *Tgfbr1* (heart, neuromesodermal progenitors) some of which would also be hit by a *Cdx2-cre* driver without temporal control. The *Cdx2-cre^{ERT}* mice used in this work had already been generated in Jurberg et al, 2013 and used in Arias et al, 2019, (doi: 10.1016/j.devcel.2018.12.004) and de Lemos et al, 2022 (doi: 10.1242/dev.198812), so we were confident with its activity. We now also added a figure in which we used the Rosa26R reporter line to show the activity of the *Cdx2-cre^{ERT}* line when induced by tamoxifen using a scheme like that employed for the *Tgfbr1* conditional inactivation (supplementary figure 1E). We also showed that our tamoxifen delivery scheme allows efficient deletion in the tissues relevant for our study using this cre driver (supplementary Figure 1F).

Following the reviewer suggestion, we had also tried the experiment with the *Isl1-cre* driver. However, the embryos died too early (around E10.5) most likely due to *Tgfbr1* inactivation in the heart (where *Isl1* is also active).

REVIEWER COMMENTS

Reviewer #1 (Remarks to the Author):

In the text of the manuscript, the authors state, "The essential role of Tgfb1 signaling in the induction of both hindlimb and GT primordia...prompted us to evaluate whether it also played a role at later stages of hindlimb and/or GT development." However, the response to reviewers' acknowledges that this was an accidental discovery after the authors initially set out to explore the role of Tgfb1 in spinal cord development. Why not say that? That is an interesting story and a good example of the real value of serendipity in science. It also better explains certain details about the experiments (e.g. the choice of Cdx2-CreERT, tamoxifen induction at E7.25) and connects this article better to the first paragraph of the introduction, which highlights the lab's longstanding interest in axial structures.

This would also better highlight what I think is the most interesting aspect of this manuscript – I have always found it fascinating that the limbs and genital tubercle share so much of their signaling and transcription factor networks, but I more or less attributed that to their shared embryonic origins and the fact they are both appendages. I never would have expected that a single genetic manipulation could transform one into the other.

The evaluation of limb patterning has been improved in this revision and gives better evidence for the gene expression patterns that accompany the transformation of pericloacal mesoderm into the extra limb. However, I think the authors have gone a step too far in this revision by invoking "differential segregation" and an "inability to mix". To me this implies a difference in adhesion molecule expression, which has not been demonstrated. I think it is fine to speculate in the discussion IF it's more clear that this is speculation AND if other equally plausible explanations are considered. For example, Figures 3A and 3B and 4A and 4B suggest discontinuities in patterning gene expression prior to morphogenesis of two distinct limbs. This outgrowth isn't being patterned as a single larger limb, and so perhaps the skeletal segregation is the outcome of that discontinuous earlier pattern.

Finally, I think the patterns of chromatin accessibility are very interesting and suggest the extra hindlimb has a mixed molecular identity. The peaks selected for transgenesis lend further support to the hypothesis that loss of Tgfb1 alters the accessibility of putative cis-regulatory sequences that distinguish the genital tubercle from the limb bud. But given the ATAC-Seq signature is a snapshot of network propagation (primary accessibility changes downstream of Tgfb1 loss of function and those due to all of the secondary, tertiary, etc, effects), I'm not sure the absence of Smad binding site enrichment in that complex set of effects is evidence that Smad DOESN'T function downstream of Tgfb1 in this system. In other words, if the primary downstream cause(s) weren't diluted by networked effects, might you see a Smad binding site enrichment? Instead the authors can state "there isn't obvious evidence that this is due to Smad" rather than "it isn't due to Smad".

Reviewer #2 (Remarks to the Author):

I appreciate the authors' efforts in addressing many of the concerns raised in my initial review. However, after carefully reviewing the revised manuscript, I find that several points still require clarification:

- As previously noted in my initial review, the discrepancy in timing between the early removal of *Tgfb1* and the subsequent ATAC-seq analysis, performed around 4 days later (E11.5), raises questions about the directness of the observed effects. moderating the claim about a distinct mode of action for *Tgfb1* and ensuring a proper acknowledgment of the potential for an indirect effect established over time. The onset of the phenotype in the hindgut and urogenital tract likely occurs very early. In addition, there is a possibility that Smad-dependent signaling remains relevant through interactions with unknown transcription factors.
- Further clarification is needed regarding the extralimb phenotype, particularly in relation to anterior-posterior (AP) patterning. Do the "normal" hindlimbs show AP polarity? How is the AP pattern in the extralimbs? Is there a digit 1? This information is crucial for an understanding of the pattern.
- Given the authors' examination of a minimum of three specimens per stage and gene expression, they should explicitly mention whether they have observed variability in the expression domains.
- The hypothesis proposing the separation of hindlimbs due to different cell affinities, driven by distinct enhancer sets, lacks sufficient support from the current data. It would be necessary to either substantiate this claim by identifying differentially expressed cell-affinity genes and perform functional studies. Otherwise, this remains purely speculative and should be tempered.
- Concerning the *Lef1* control of *Isl1*, the authors may want to explore potential integration with the established interaction between *bCat/islet1* (see doi:10.1242/dev.073056 and doi:10.1242/dev.065359). Is the genotype in Supplementary Figure 4 the same as in Figure 6D? It seems that the reported expression throughout the embryo is only visible in Figure 6D.
- When referencing the sirenornelia models, please include the *Bmp7;Shh* mutant as well (PMID: 23028704) for comprehensive coverage.

Reviewer #3 (Remarks to the Author):

The revised version replied to the reviewer's comments on the next points;

Marker and regulatory region analysis and informatics

Cre activity of *Cdx* and *Tm* conditions

Isl1 Cre possibility

Previous works citation including sirenornelia

Corrections on pioneer activity

The concept of organ anlage fate change is rather hard to "prove" unlike the case for cell level experiments.

I agree the fate change “possibility” was basically shown by the current series of data set in the MS.

The MS will appeal to NC readers particularly with the duplicated limb phenotype.

REVIEWER COMMENTS

Reviewer #1 (Remarks to the Author):

In the text of the manuscript, the authors state, "The essential role of Tgfbr1 signaling in the induction of both hindlimb and GT primordia...prompted us to evaluate whether it also played a role at later stages of hindlimb and/or GT development." However, the response to reviewers' acknowledges that this was an accidental discovery after the authors initially set out to explore the role of Tgfbr1 in spinal cord development. Why not say that? That is an interesting story and a good example of the real value of serendipity in science. It also better explains certain details about the experiments (e.g. the choice of Cdx2-CreERT, tamoxifen induction at E7.25) and connects this article better to the first paragraph of the introduction, which highlights the lab's longstanding interest in axial structures.

This would also better highlight what I think is the most interesting aspect of this manuscript – I have always found it fascinating that the limbs and genital tubercle share so much of their signaling and transcription factor networks, but I more or less attributed that to their shared embryonic origins and the fact they are both appendages. I never would have expected that a single genetic manipulation could transform one into the other.

REPLY

We agree that admitting that our results were unexpected and diverged from initial question make the introduction more genuine and fits better to justify the model. We changed the last section of the introduction to fit better the original reason for making the conditional mutants and the unexpected finding that then was the basis of most of the work described in this manuscript (lines 115-126).

The evaluation of limb patterning has been improved in this revision and gives better evidence for the gene expression patterns that accompany the transformation of pericloacal mesoderm into the extra limb. However, I think the authors have gone a step too far in this revision by invoking "differential segregation" and an "inability to mix". To me this implies a difference in adhesion molecule expression, which has not been demonstrated. I think it is fine to speculate in the discussion IF it's more clear that this is speculation AND if other equally plausible explanations are considered. For example, Figures 3A and 3B and 4A and 4B suggest discontinuities in patterning gene expression prior to morphogenesis of two distinct limbs. This outgrowth isn't being patterned as a single larger limb, and so perhaps the skeletal segregation is the outcome of that discontinuous earlier pattern.

REPLY

We agree that based on our data the different cell affinity between the hindlimb and pericloacal tissues is just one of the possible hypotheses to explain the generation of two independent hindlimb structures, and that other explanations are also possible based on the discontinuity of some gene expression markers at early stages of formation of the hindlimb structures in from the initial bud. We changed the relevant section of the discussion accordingly (lines 511 – 535).

Finally, I think the patterns of chromatin accessibility are very interesting and suggest the extra hindlimb has a mixed molecular identity. The peaks selected for transgenesis lend further support to the hypothesis that loss of Tgfrb1 alters the accessibility of putative cis-regulatory sequences that distinguish the genital tubercle from the limb bud. But given the ATAC-Seq signature is a snapshot of network propagation (primary accessibility changes downstream of Tgfrb1 loss of function and those due to all of the secondary, tertiary, etc, effects), I'm not sure the absence of Smad binding site enrichment in that complex set of effects is evidence that Smad DOESN'T function downstream of Tgfrb1 in this system. In other words, if the primary downstream cause(s) weren't diluted by networked effects, might you see a Smad binding site enrichment? Instead the authors can state "there isn't obvious evidence that this is due to Smad" rather than "it isn't due to Smad".

REPLY

We agree with the reviewer and edited the corresponding section in the discussion accordingly (lines 564 – 578).

Reviewer #2 (Remarks to the Author):

I appreciate the authors' efforts in addressing many of the concerns raised in my initial review. However, after carefully reviewing the revised manuscript, I find that several points still require clarification:

- As previously noted in my initial review, the discrepancy in timing between the early removal of Tgfrb1 and the subsequent ATAC-seq analysis, performed around 4 days later (E11.5), raises questions about the directness of the observed effects. moderating the claim about a distinct mode of action for Tgfrb1 and ensuring a proper acknowledgment of the potential for an indirect effect established over time. The onset of the phenotype in the hindgut and urogenital tract likely occurs very early. In addition, there is a possibility that Smad-dependent signaling remains relevant through interactions with unknown transcription factors.

REPLY

Similar concern was raised by the Reviewer #1 and we have now addressed it in the discussion, lines 564 – 578.

- Further clarification is needed regarding the extralimb phenotype, particularly in relation to anterior-posterior (AP) patterning. Do the "normal" hindlimbs show AP polarity? How is the AP pattern in the extralimbs? Is there a digit 1? This information is crucial for an understanding of the pattern.

REPLY

We added segmentation of the "normal" hindlimb in the mutant in the supplementary fig. 2. Both limbs form 5 digits with altered morphology; therefore, it is hard to confirm each digit's identity with certainty. The normal hindlimb shows anterior Pax9 expression, so it has some level of AP polarity. Regarding the digit 1, it seems like it is present in the analyzed late-stage specimen (sup. Fig. 2B-C') based on the number of phalanges.

- Given the authors' examination of a minimum of three specimens per stage and gene expression, they should explicitly mention whether they have observed variability in the expression domains.

REPLY

Expression patterns showed a high degree of consistency, especially considering a tamoxifen-driven conditional deletion. The correspondent statement was added to the methods section lines 851-854.

- The hypothesis proposing the separation of hindlimbs due to different cell affinities, driven by distinct enhancer sets, lacks sufficient support from the current data. It would be necessary to either substantiate this claim by identifying differentially expressed cell-affinity genes and perform functional studies. Otherwise, this remains purely speculative and should be tempered.

REPLY

Was addressed this in the answer to reviewer #1 in lines 511 – 535.

- Concerning the *Lef1* control of *Isl1*, the authors may want to explore potential integration with the established interaction between *bCat/islet1* (see [doi:10.1242/dev.073056](https://doi.org/10.1242/dev.073056) and [doi:10.1242/dev.065359](https://doi.org/10.1242/dev.065359)).

REPLY

Indeed, there is reported interaction between the *Isl1* and *bCat* in the context of hindlimb development. Particularly, *Isl1* is required for nuclear *bCat* accumulation in the context of the hindlimb induction. We did not introduce this connection because in our case the interaction is reversed: we show that Wnt signaling regulates potential *Isl1* enhancer. Furthermore, CR-*Isl1* is a potential *Isl1*-regulatory region, although likely involved in GT development, we can't be sure that it indeed regulates

Isl1 expression. Altogether, although we acknowledge that there might be some connection with the previously published data, we think that in case of our study drawing connection might come out highly speculative.

Is the genotype in Supplementary Figure 4 the same as in Figure 6D? It seems that the reported expression throughout the embryo is only visible in Figure 6D.

REPLY

The embryos to which the reviewer refers to have the same genotype (*CR-Isl1^{ΔA}*). In the embryo shown in supplementary Figure 4E there is also some faint expression (e.g. in the somites, the proximal forelimb and hindbrain). However, the point here is that in the absence of the Lef1 and Gata sites, this enhancer loses specificity and, when some expression is detected, it might be related to the insertion site (this is why the two specimens show different expression patterns) but not to the intrinsic activity of the enhancer. We introduced a note in the results (lines 453-455).

- When referencing the sirenomelia models, please include the Bmp7;Shh mutant as well (PMID: 23028704) for comprehensive coverage.

REPLY

Reference added.

REVIEWERS' COMMENTS

Reviewer #1 (Remarks to the Author):

The authors have addressed each of my points, and I congratulate them on an interesting and exciting study. A final minor point in line 23: please change to 'in addition to confirming'.

Reviewer #2 (Remarks to the Author):

The authors have addressed my requests

Through the revision process, the paper has significantly improved, and I find it acceptable for publication.